# WHEN IS A CONVOLUTIONAL FILTER EASY TO LEARN?

**Simon S. Du**
Carnegie Mellon University
ssdu@cs.cmu.edu

**Jason D. Lee**
University of Southern California
jasonlee@marshall.usc.edu

**Yuandong Tian**
Facebook AI Research
yuandong@fb.com

## ABSTRACT

We analyze the convergence of (stochastic) gradient descent algorithm for learning a convolutional filter with Rectified Linear Unit (ReLU) activation function. Our analysis does not rely on any specific form of the input distribution and our proofs only use the definition of ReLU, in contrast with previous works that are restricted to standard Gaussian input. We show that (stochastic) gradient descent with random initialization can learn the convolutional filter in polynomial time and the convergence rate depends on the smoothness of the input distribution and the closeness of patches. To the best of our knowledge, this is the first recovery guarantee of gradient-based algorithms for convolutional filter on non-Gaussian input distributions. Our theory also justifies the two-stage learning rate strategy in deep neural networks. While our focus is theoretical, we also present experiments that justify our theoretical findings.

## 1 INTRODUCTION

Deep convolutional neural networks (CNN) have achieved the state-of-the-art performance in many applications such as computer vision (Krizhevsky et al., 2012), natural language processing (Dauphin et al., 2016) and reinforcement learning applied in classic games like Go (Silver et al., 2016). Despite the highly non-convex nature of the objective function, simple first-order algorithms like stochastic gradient descent and its variants often train such networks successfully. On the other hand, the success of convolutional neural network remains elusive from an optimization perspective.

When the input distribution is not constrained, existing results are mostly negative, such as hardness of learning a 3-node neural network (Blum & Rivest, 1989) or a non-overlap convolutional filter (Brutzkus & Globerson, 2017). Recently, Shamir (2016) showed learning a simple one-layer fully connected neural network is hard for some specific input distributions.

These negative results suggest that, in order to explain the empirical success of SGD for learning neural networks, stronger assumptions on the input distribution are needed. Recently, a line of research (Tian, 2017; Brutzkus & Globerson, 2017; Li & Yuan, 2017; Soltanolkotabi, 2017; Zhong et al., 2017) assumed the input distribution be *standard Gaussian* $N(0, \mathbf{I})$ and showed (stochastic) gradient descent is able to recover neural networks with ReLU activation in polynomial time.

One major issue of these analysis is that they rely on specialized analytic properties of the Gaussian distribution (c.f. Section 1.1) and thus *cannot* be generalized to the non-Gaussian case, in which real-world distributions fall into. For general input distributions, new techniques are needed.

In this paper we consider a simple architecture: a convolution layer, followed by a ReLU activation function, and then average pooling. Formally, we let $\mathbf{x} \in \mathbb{R}^d$ be an input sample, e.g., an image, we generate $k$ patches from $\mathbf{x}$, each with size $p$: $\mathbf{Z} \in \mathbb{R}^{p \times k}$ where the $i$-th column is the $i$-th patch generated by some known function $\mathbf{Z}_i = \mathbf{Z}_i(\mathbf{x})$. For a filter with size 2 and stride 1, $\mathbf{Z}_i(\mathbf{x})$ is the $i$-th and $(i+1)$-th pixels. Since for convolutional filters, we only need to focus on the patches instead of the input, in the following definitions and theorems, we will refer $\mathbf{Z}$ as input and let $\mathcal{Z}$ as the distribution of $\mathbf{Z}$: ($\sigma(x) = \max(x, 0)$ is the ReLU activation function)

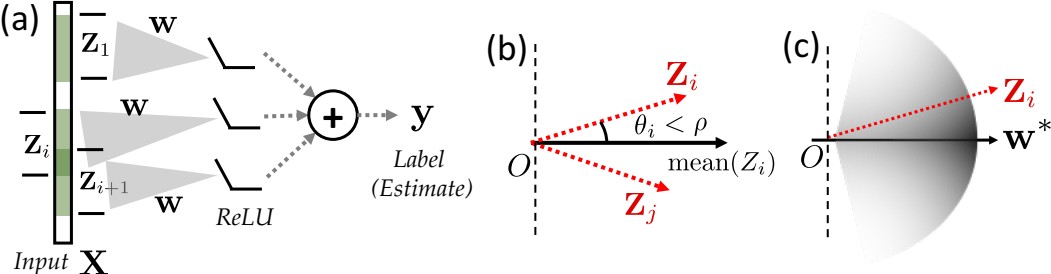

Figure 1: **(a)** Architecture of the network we are considering. Given input $X$, we extract its patches $\{Z_i\}$ and send them to a shared weight vector $\mathbf{w}$. The outputs are then sent to ReLU and then summed to yield the final label (and its estimation). **(b)-(c)** Two conditions we proposed for convergence. We want the data to be (b) highly correlated and (c) concentrated more on the direction aligned with the ground truth vector $\mathbf{w}^*$.

$$f(\mathbf{w}, \mathbf{Z}) = \frac{1}{k} \sum_{i=1}^{k} \sigma\left(\mathbf{w}^{\top} \mathbf{Z}_i\right). \tag{1}$$

See Figure 1 (a) for a graphical illustration. Such architectures have been used as the first layer of many works in computer vision (Lin et al., 2013; Milletari et al., 2016). We address the realizable case, where training data are generated from (1) with some unknown teacher parameter $\mathbf{w}_*$ under input distribution $\mathcal{Z}$. Consider the $\ell_2$ loss $\ell(\mathbf{w}, \mathbf{Z}) = \frac{1}{2}(f(\mathbf{w}, \mathbf{Z}) - f(\mathbf{w}_*, \mathbf{Z}))^2$. We learn by (stochastic) gradient descent, i.e.,

$$\mathbf{w}_{t+1} = \mathbf{w}_t - \eta_t g(\mathbf{w}_t) \tag{2}$$

where $\eta_t$ is the step size which may change over time and $g(\mathbf{w}_t)$ is a random function where its expectation equals to the population gradient $\mathbb{E}[g(\mathbf{w})] = \mathbb{E}_{\mathbf{Z} \sim \mathcal{Z}}[\nabla \ell(\mathbf{w}, \mathbf{Z})]$. The goal of our analysis is to understand the conditions where $\mathbf{w} \to \mathbf{w}_*$, if $\mathbf{w}$ is optimized under (stochastic) gradient descent.

In this setup, our main contributions are as follows:

- **Learnability of Filters:** We show if the input patches are highly correlated (Section 3), i.e., $\theta(\mathbf{Z}_i, \mathbf{Z}_j) \leq \rho$ for some small $\rho > 0$, then gradient descent and stochastic gradient descent with random initialization recovers the filter in polynomial time.[1] Furthermore, strong correlations imply faster convergence. To the best of our knowledge, this is the first recovery guarantee of randomly initialized gradient-based algorithms for learning filters (even for the simplest one-layer one-neuron network) on non-Gaussian input distribution, answering an open problem in (Tian, 2017).

- **Distribution-Aware Convergence Rate**. We formally establish the connection between the smoothness of the input distribution and the convergence rate for filter weights recovery where the smoothness in our paper is defined as the ratio between the largest and the least eigenvalues of the second moment of the activation region (Section 2). We show that a smoother input distribution leads to faster convergence, and Gaussian distribution is a special case that leads to the tightest bound. This theoretical finding also justifies the two-stage learning rate strategy proposed by (He et al., 2016; Szegedy et al., 2017) if the step size is allowed to change over time.

## 1.1 RELATED WORKS

In recent years, theorists have tried to explain the success of deep learning from different perspectives. From optimization point of view, optimizing neural network is a non-convex optimization

---

[1]Note since in this paper we focus on continuous distribution over $\mathbf{Z}$, our results do not conflict with previous negative results(Blum & Rivest, 1989; Brutzkus & Globerson, 2017) whose constructions rely on discrete distributions.

problem. Pioneered by Ge et al. (2015), a class of non-convex optimization problems that satisfy strict saddle property can be optimized by perturbed (stochastic) gradient descent in polynomial time (Jin et al., 2017).[2] This motivates the research of studying the landscape of neural networks (Soltanolkotabi et al., 2017; Kawaguchi, 2016; Choromanska et al., 2015; Hardt & Ma, 2016; Haeffele & Vidal, 2015; Mei et al., 2016; Freeman & Bruna, 2016; Safran & Shamir, 2016; Zhou & Feng, 2017; Nguyen & Hein, 2017) However, these results cannot be directly applied to analyzing the convergence of gradient-based methods for ReLU activated neural networks.

From learning theory point of view, it is well known that training a neural network is hard in the worst cases (Blum & Rivest, 1989; Livni et al., 2014; Šíma, 2002; Shalev-Shwartz et al., 2017a;b) and recently, Shamir (2016) showed either "niceness" of the target function or of the input distribution alone is sufficient for optimization algorithms used in practice to succeed. With some additional assumptions, many works tried to design algorithms that provably learn a neural network with polynomial time and sample complexity (Goel et al., 2016; Zhang et al., 2016; 2015; Sedghi & Anandkumar, 2014; Janzamin et al., 2015; Gautier et al., 2016; Goel & Klivans, 2017). However, these algorithms are tailored for certain architecture and cannot explain why (stochastic) gradient based optimization algorithms work well in practice.

Focusing on gradient-based algorithms, a line of research analyzed the behavior of (stochastic) gradient descent for *Gaussian* input distribution. Tian (2017) showed population gradient descent is able to find the true weight vector with random initialization for one-layer one-neuron model. Brutzkus & Globerson (2017) showed population gradient descent recovers the true weights of a convolution filter with non-overlapping input in polynomial time. Li & Yuan (2017) showed SGD can recover the true weights of a one-layer ResNet model with ReLU activation under the assumption that the spectral norm of the true weights is bounded by a small constant. All the methods use explicit formulas for Gaussian input, which enable them to apply trigonometric inequalities to derive the convergence. With the same Gaussian assumption, Soltanolkotabi (2017) shows that the true weights can be exactly recovered by projected gradient descent with enough samples in linear time, if the number of inputs is less than the dimension of the weights.

Other approaches combine tensor approaches with assumptions of input distribution. Zhong et al. (2017) proved that with sufficiently good initialization, which can be implemented by tensor method, gradient descent can find the true weights of a 3-layer fully connected neural network. However, their approach works with known input distributions. Soltanolkotabi (2017) used Gaussian width (c.f. Definition 2.2 of (Soltanolkotabi, 2017)) for concentrations and his approach cannot be directly extended to learning a convolutional filter.

In this paper, we adopt a different approach that only relies on the definition of ReLU. We show as long as the input distribution satisfies weak smoothness assumptions, we are able to find the true weights by SGD in polynomial time. Using our conclusions, we can justify the effectiveness of large amounts of data (which may eliminate saddle points), two-stage and adaptive learning rates used by He et al. (2016); Szegedy et al. (2017), etc.

## 1.2 ORGANIZATION

This paper is organized as follows. In Section 2, we analyze the simplest one-layer one-neuron model where we state our key observation and establish the connection between smoothness and convergence rate. In Section 3, we discuss the performance of (stochastic) gradient descent for learning a convolutional filter. We provide empirical illustrations in Section 4 and conclude in Section 5. We place most of our detailed proofs in the Appendix.

## 1.3 NOTATIONS

Let $\|\cdot\|_2$ denote the Euclidean norm of a finite-dimensional vector. For a matrix $\mathbf{A}$, we use $\lambda_{\max}(\mathbf{A})$ to denote its largest singular value and $\lambda_{\min}(\mathbf{A})$ its smallest singular value. Note if $\mathbf{A}$ is a positive semidefinite matrix, $\lambda_{\max}(\mathbf{A})$ and $\lambda_{\min}(\mathbf{A})$ represent the largest and smallest eigenvalues of $\mathbf{A}$, respectively. Let $O(\cdot)$ and $\Theta(\cdot)$ denote the standard Big-O and Big-Theta notations that hide absolute

---

[2]Gradient descent is not guaranteed to converge to a local minima in polynomial time (Du et al., 2017; Lee et al., 2016).

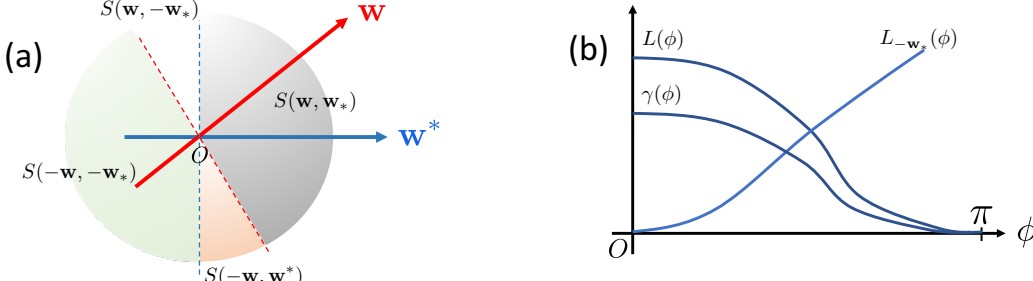

Figure 2: **(a)** The four regions considered in our analysis. **(b)** Illustration of $L(\phi), \gamma(\phi)$ and $L_{-\mathbf{w}_*}(\phi)$ defined in Definition 2.1 and Assumption 2.1.

constants. We assume the gradient function is uniformly bounded, i.e., There exists $B > 0$ such that $\|g(\mathbf{w})\|_2 \leq B$. This condition is satisfied as long as patches, $\mathbf{w}$ and noise are all bounded.

## 2    WARM UP: ANALYZING ONE-LAYER ONE-NEURON MODEL

Before diving into the convolutional filter, we first analyze the special case for $k = 1$, which is equivalent to the one-layer one-neuron architecture. The analysis in this simple case will give us insights for the fully general case. For the ease of presentation, we define following two events and corresponding second moments

$$S(\mathbf{w}, \mathbf{w}_*) = \left\{ \mathbf{Z} : \mathbf{w}^\top \mathbf{Z} \geq 0, \mathbf{w}_*^\top \mathbf{Z} \geq 0 \right\}, \quad S(\mathbf{w}, -\mathbf{w}_*) = \left\{ \mathbf{Z} : \mathbf{w}^\top \mathbf{Z} \geq 0, \mathbf{w}_*^\top \mathbf{Z} \leq 0 \right\}, \quad (3)$$

$$\mathbf{A}_{\mathbf{w}, \mathbf{w}_*} = \mathbb{E}\left[ \mathbf{Z}\mathbf{Z}^\top \mathbb{I}\{S(\mathbf{w}, \mathbf{w}_*)\} \right], \quad \mathbf{A}_{\mathbf{w}, -\mathbf{w}_*} = \mathbb{E}\left[ \mathbf{Z}\mathbf{Z}^\top \mathbb{I}\{S(\mathbf{w}, -\mathbf{w}_*)\} \right].$$

where $\mathbb{I}\{\cdot\}$ is the indicator function. Intuitively, $S(\mathbf{w}, \mathbf{w}_*)$ is the joint activation region of $\mathbf{w}$ and $\mathbf{w}_*$ and $S(\mathbf{w}, -\mathbf{w}_*)$ is the joint activation region of $\mathbf{w}$ and $-\mathbf{w}_*$. See Figure 2 (a) for the graphical illustration. With some simple algebra we can derive the population gradient.

$$\mathbb{E}\left[ \nabla \ell(\mathbf{w}, \mathbf{Z}) \right] = \mathbf{A}_{\mathbf{w}, \mathbf{w}_*}(\mathbf{w} - \mathbf{w}_*) + \mathbf{A}_{\mathbf{w}, -\mathbf{w}_*}\mathbf{w}.$$

One key observation is we can write the inner product $\langle \nabla_\mathbf{w} \ell(\mathbf{w}), \mathbf{w} - \mathbf{w}_* \rangle$ as the sum of two non-negative terms (c.f. Lemma A.1). This observation directly leads to the following Theorem 2.1.

**Theorem 2.1.** *Suppose for any $\mathbf{w}_1, \mathbf{w}_2$ with $\theta(\mathbf{w}_1, \mathbf{w}_2) < \pi$, $\mathbb{E}\left[ \mathbf{Z}\mathbf{Z}^\top \mathbb{I}\{S(\mathbf{w}, \mathbf{w}_*)\} \right] \succ 0$ and the initialization $\mathbf{w}_0$ satisfies $\ell(\mathbf{w}_0) < \ell(\mathbf{0})$ then gradient descent algorithm recovers $\mathbf{w}_*$.*

The first assumption is about the non-degeneracy of input distribution. For $\theta(\mathbf{w}_1, \mathbf{w}_2) < \pi$, one case that the assumption fails is that the input distribution is supported on a low-dimensional space, or degenerated. The second assumption on the initialization is to ensure that gradient descent does not converge to $\mathbf{w} = \mathbf{0}$, at which the gradient is undefined. This is a general convergence theorem that holds for a wide class of input distribution and initialization points. In particular, it includes Theorem 6 of (Tian, 2017) as a special case. If the input distribution is degenerate, i.e., there are holes in the input space, the gradient descent may stuck around saddle points and we believe more data are needed to facilitate the optimization procedure This is also consistent with empirical evidence in which more data are helpful for optimization.

### 2.1    CONVERGENCE RATE OF ONE-LAYER ONE-NEURON MODEL

In the previous section we showed if the distribution is regular and the weights are initialized appropriately, gradient descent recovers the true weights when it converges. In practice we also want to know how many iterations are needed. To characterize the convergence rate, we need some quantitative assumptions. We note that different set of assumptions will lead to a different rate and ours is only one possible choice. In this paper we use the following quantities.

**Definition 2.1** (The Largest/Smallest eigenvalue Values of the Second Moment on Intersection of two Half Spaces). *For $\phi \in [0, \pi]$, define*

$$\gamma(\phi) = \min_{\mathbf{w}:\angle\mathbf{w},\mathbf{w}_*=\phi} \lambda_{\min}(\mathbf{A}_{\mathbf{w},\mathbf{w}_*}), \quad L(\phi) = \max_{\mathbf{w}:\angle\mathbf{w},\mathbf{w}_*=\phi} \lambda_{\max}(\mathbf{A}_{\mathbf{w},\mathbf{w}_*}),$$

These two conditions quantitatively characterize the angular smoothness of the input distribution. For a given angle $\phi$, if the difference between $\gamma(\phi)$ and $L(\phi)$ is large then there is one direction has large probability mass and one direction has small probability mass, meaning the input distribution is not smooth. On the other hand, if $\gamma(\phi)$ and $L(\phi)$ are close, then all directions have similar probability mass, which means the input distribution is smooth. The smoothest input distributions are rotationally invariant distributions (e.g. standard Gaussian) which have $\gamma(\phi) = L(\phi)$. For analogy, we can think of $L(\phi)$ as Lipschitz constant of the gradient and $\gamma(\phi)$ as the strong convexity parameter in the optimization literature but here we also allow they change with the angle. Also observe that when $\phi = \pi$, $\gamma(\phi) = L(\phi) = 0$ because the intersection has measure 0 and both $\gamma(\phi)$ and $L(\phi)$ are monotonically decreasing.

Our next assumption is on the growth of $\mathbf{A}_{\mathbf{w},-\mathbf{w}_*}$. Note that when $\theta(\mathbf{w}, \mathbf{w}_*) = 0$, then $\mathbf{A}_{\mathbf{w},-\mathbf{w}_*} = \mathbf{0}$ because the intersection between $\mathbf{w}$ and $-\mathbf{w}_*$ has 0 measure. Also, $\mathbf{A}_{\mathbf{w},-\mathbf{w}_*}$ grows as the angle between $\mathbf{w}$ and $\mathbf{w}_*$ becomes larger.

In the following, we assume the operator norm of $\mathbf{A}_{\mathbf{w},-\mathbf{w}_*}$ increases smoothly with respect to the angle. The intuition is that as long as input distribution bounded probability density with respect to the angle, the operator norm of $\mathbf{A}_{\mathbf{w},-\mathbf{w}_*}$ is bounded. We show in Theorem A.1 that $\beta = 1$ for rotational invariant distribution and in Theorem A.2 that $\beta = p$ for standard Gaussian distribution.

**Assumption 2.1.** *We assume there exists $\beta > 0$ that for $0 \leq \phi \leq \pi/2$, $L_{-w_*}(\phi) \triangleq \max_{\mathbf{w},\theta(\mathbf{w},\mathbf{w}_*)\leq\phi} \lambda_{\max}(\mathbf{A}_{\mathbf{w},-\mathbf{w}_*}) \leq \beta\phi$.*

Now we are ready to state the convergence rate.

**Theorem 2.2.** *Suppose the initialization $\mathbf{w}_0$ satisfies $\|\mathbf{w}_0 - \mathbf{w}_*\|_2 < \|\mathbf{w}_*\|_2$. Denote $\phi_t = \arcsin\left(\frac{\|\mathbf{w}_t - \mathbf{w}_*\|_2}{\|\mathbf{w}_*\|_2}\right)$ then if step size is set as $0 \leq \eta_t \leq \min_{0\leq\phi\leq\phi_t} \frac{\gamma(\phi)}{2(L(\phi)+4\beta)^2}$, we have for $t = 1, 2, \ldots$*

$$\|\mathbf{w}_{t+1} - \mathbf{w}_*\|_2^2 \leq \left(1 - \frac{\eta_t \gamma(\phi_t)}{2}\right) \|\mathbf{w}_t - \mathbf{w}_*\|_2^2.$$

Note both $\gamma(\phi)$ and $L(\phi)$ increases as $\phi$ decreases so we can choose a constant step size $\eta_t = \Theta\left(\frac{\gamma(\phi_0)}{(L(0)+\beta)^2}\right)$. This theorem implies that we can find the $\epsilon$-close solution of $\mathbf{w}_*$ in $O\left(\frac{(L(0)+\beta)^2}{\gamma^2(\phi_0)} \log\left(\frac{1}{\epsilon}\right)\right)$ iterations. It also suggests a direct relation between the smoothness of the distribution and the convergence rate. For smooth distribution where $\gamma(\phi)$ and $L(\phi)$ are close and $\beta$ is small then $\frac{(L(0)+\beta)^2}{\gamma^2(\phi_0)}$ is relatively small and we need fewer iterations. On the other hand, if $L(\phi)$ or $\beta$ is much larger than $\gamma(\phi)$, we will need more iterations. We verify this intuition in Section 4.

If we are able to choose the step sizes adaptively $\eta_t = \Theta\left(\frac{\gamma(\phi_t)}{(L(\phi_t)+\beta)^2}\right)$, like using methods proposed by Lin & Xiao (2014), we may improve the computational complexity to $O\left(\max_{\phi\leq\phi_0} \frac{(L(\phi)+\beta)^2}{\gamma^2(\phi)} \log\left(\frac{1}{\epsilon}\right)\right)$. This justifies the use of two-stage learning rate strategy proposed by He et al. (2016); Szegedy et al. (2017) where at the beginning we need to choose learning to be small because $\frac{\gamma(\phi_0)}{2(L(\phi_0)+2\beta)^2}$ is small and later we can choose a large learning rate because as the angle between $\mathbf{w}_t$ and $\mathbf{w}_*$ becomes smaller, $\frac{\gamma(\phi_t)}{2(L(\phi_t)+2\beta)^2}$ becomes bigger.

The theorem requires the initialization satisfying $\|\mathbf{w}_0 - \mathbf{w}_*\|_2 < \|\mathbf{w}_*\|_2$, which can be achieved by random initialization with constant success probability. See Section 3.2 for a detailed discussion.

## 3 MAIN RESULTS FOR LEARNING A CONVOLUTIONAL FILTER

In this section we generalize ideas from the previous section to analyze the convolutional filter. First, for given $\mathbf{w}$ and $\mathbf{w}_*$ we define four events that divide the input space of each patch $\mathbf{Z}_i$. Each event

corresponds to a different activation region induced by $\mathbf{w}$ and $\mathbf{w}_*$, similar to (3).

$$S(\mathbf{w}, \mathbf{w}_*)_i = \left\{ \mathbf{Z}_i : \mathbf{w}^\top \mathbf{Z}_i \geq 0, \mathbf{w}_*^\top \mathbf{Z}_i \geq 0 \right\}, \quad S(\mathbf{w}, -\mathbf{w}_*)_i = \left\{ \mathbf{Z}_i : \mathbf{w}^\top \mathbf{Z}_i \geq 0, \mathbf{w}_*^\top \mathbf{Z}_i \leq 0 \right\},$$
$$S(-\mathbf{w}, -\mathbf{w}_*)_i = \left\{ \mathbf{Z}_i : \mathbf{w}^\top \mathbf{Z}_i \leq 0, \mathbf{w}_*^\top \mathbf{Z}_i \leq 0 \right\}, \quad S(-\mathbf{w}, \mathbf{w}_*)_i = \left\{ \mathbf{Z}_i : \mathbf{w}^\top \mathbf{Z}_i \leq 0, \mathbf{w}_*^\top \mathbf{Z}_i \geq 0 \right\}.$$

Please check Figure 2 (a) again for illustration. For the ease of presentation we also define the average over all patches in each region

$$\mathbf{Z}_{S(\mathbf{w}, \mathbf{w}_*)} = \frac{1}{k} \sum_{i=1}^{k} \mathbf{Z}_i \mathbb{I} \left\{ S(\mathbf{w}, \mathbf{w}_*)_i \right\}, \mathbf{Z}_{S(\mathbf{w}, -\mathbf{w}_*)} = \frac{1}{k} \sum_{i=1}^{k} \mathbf{Z}_i \mathbb{I} \left\{ S(\mathbf{w}, -\mathbf{w}_*)_i \right\},$$

$$\mathbf{Z}_{S(-\mathbf{w}, \mathbf{w}_*)} = \frac{1}{k} \sum_{i=1}^{k} \mathbf{Z}_i \mathbb{I} \left\{ S(-\mathbf{w}, \mathbf{w}_*)_i \right\}.$$

Next, we generalize the smoothness conditions analogue to Definition 2.1 and Assumption 2.1. Here the smoothness is defined over the average of patches.

**Assumption 3.1.** *For $\phi \in [0, \pi]$, define*

$$\gamma(\phi) = \min_{\mathbf{w}:\theta(\mathbf{w}, \mathbf{w}_*)=\phi} \lambda_{\min} \left( \mathbb{E} \left[ \mathbf{Z}_{S(\mathbf{w}, \mathbf{w}_*)} \mathbf{Z}_{S(\mathbf{w}, \mathbf{w}_*)}^\top \right] \right),$$

$$L(\phi) = \max_{\mathbf{w}:\theta(\mathbf{w}, \mathbf{w}_*)=\phi} \lambda_{\max} \left( \mathbb{E} \left[ \mathbf{Z}_{S(\mathbf{w}, \mathbf{w}_*)} \mathbf{Z}_{S(\mathbf{w}, \mathbf{w}_*)}^\top \right] \right). \tag{4}$$

*We assume for all $0 \leq \phi \leq \pi/2$, $\max_{\mathbf{w}:\theta(\mathbf{w}, \mathbf{w}_*)=\phi} \lambda_{\max} \left( \mathbb{E} \left[ \mathbf{Z}_{S(\mathbf{w}, -\mathbf{w}_*)} \mathbf{Z}_{S(\mathbf{w}, -\mathbf{w}_*)}^\top \right] \right) \leq \beta\phi$ for some $\beta > 0$.*

The main difference between the simple one-layer one-neuron network and the convolution filter is two patches may appear in different regions. For a given sample, there may exists patch $\mathbf{Z}_i$ and $\mathbf{Z}_j$ such that $\mathbf{Z}_i \in S(\mathbf{w}, \mathbf{w}_*)_i$ and $\mathbf{Z}_j \in S(\mathbf{w}, -\mathbf{w}_*)_j$ and their interaction plays an important role in the convergence of (stochastic) gradient descent. Here we assume the second moment of this interaction, i.e., cross-covariance, also grows smoothly with respect to the angle.

**Assumption 3.2.** *We assume there exists $L_{cross} > 0$ such that*

$$\max_{\mathbf{w}:\theta(\mathbf{w}, \mathbf{w}_*)\leq\phi} \lambda_{\max} \left( \mathbb{E} \left[ \mathbf{Z}_{S(\mathbf{w}, \mathbf{w}_*)} \mathbf{Z}_{S(\mathbf{w}, -\mathbf{w}_*)}^\top \right] \right) + \lambda_{\max} \left( \mathbb{E} \left[ \mathbf{Z}_{S(\mathbf{w}, \mathbf{w}_*)} \mathbf{Z}_{S(-\mathbf{w}, \mathbf{w}_*)}^\top \right] \right)$$

$$+ \lambda_{\max} \left( \mathbb{E} \left[ \mathbf{Z}_{S(\mathbf{w}, -\mathbf{w}_*)} \mathbf{Z}_{S(-\mathbf{w}, \mathbf{w}_*)}^\top \right] \right) \leq L_{cross}\phi.$$

First note if $\phi = 0$, then $\mathbf{Z}_{S(\mathbf{w}, -\mathbf{w}_*)}$ and $\mathbf{Z}_{S(-\mathbf{w}, \mathbf{w}_*)}$ has measure $0$ and this assumption models the growth of cross-covariance. Next note this $L_{cross}$ represents the closeness of patches. If $\mathbf{Z}_i$ and $\mathbf{Z}_j$ are very similar, then the joint probability density of $\mathbf{Z}_i \in S(\mathbf{w}, \mathbf{w}_*)_i$ and $\mathbf{Z}_j \in S(\mathbf{w}, -\mathbf{w}_*)_j$ is small which implies $L_{cross}$ is small. In the extreme setting, $\mathbf{Z}_1 = \ldots = \mathbf{Z}_k$, we have $L_{cross} = 0$ because in this case the events $\{\mathbf{Z}_i \in S(\mathbf{w}, \mathbf{w}_*)_i\} \cap \{\mathbf{Z}_j \in S(\mathbf{w}, -\mathbf{w}_*)_j\}$, $\{\mathbf{Z}_i \in S(\mathbf{w}, \mathbf{w}_*)_i\} \cap \{\mathbf{Z}_j \in S(-\mathbf{w}, \mathbf{w}_*)_j\}$ and $\{\mathbf{Z}_i \in S(\mathbf{w}, -\mathbf{w}_*)_i\} \cap \{\mathbf{Z}_j \in S(-\mathbf{w}, \mathbf{w}_*)_j\}$ all have measure $0$.

Now we are ready to present our result on learning a convolutional filter by gradient descent.

**Theorem 3.1.** *If the initialization satisfies $\|\mathbf{w}_0 - \mathbf{w}_*\|_2 < \|\mathbf{w}_*\|_2$ and denote $\phi_t = \arcsin\left(\frac{\|\mathbf{w}_t - \mathbf{w}_*\|_2}{\|\mathbf{w}_*\|_2}\right)$ which satisfies $\gamma(\phi_0) > 6L_{\mathrm{cross}}$. Then if we choose $\eta_t \leq \min_{0\leq\phi\leq\phi_t} \frac{\gamma(\phi)-6L_{\mathrm{cross}}}{2(L(\phi)+10L_{\mathrm{cross}}+4\beta)^2}$, we have for $t = 1, 2, \ldots$ and $\phi_t \triangleq \arcsin\left(\frac{\|\mathbf{w}_t - \mathbf{w}_*\|_2}{\|\mathbf{w}_*\|_2}\right)$*

$$\|\mathbf{w}_{t+1} - \mathbf{w}_*\|_2^2 \leq \left(1 - \frac{\eta(\gamma(\phi_t) - 6L_{\mathrm{cross}})}{2}\right) \|\mathbf{w}_t - \mathbf{w}_*\|_2^2$$

Our theorem suggests if the initialization satisfies $\gamma(\phi_0) > 6L_{\mathrm{cross}}$, we obtain linear convergence rate. In Section 3.1, we give a concrete example showing closeness of patches implies large $\gamma(\phi)$ and small $L_{\mathrm{cross}}$. Similar to Theorem 2.2, if the step size is chosen so that $\eta_t =$

$\Theta\left(\frac{\gamma(\phi_0)-6L_{\mathrm{cross}}}{\left(L_{S(\mathbf{w},\mathbf{w}_*)}(0)+10L_{\mathrm{cross}}+4\beta\right)^2}\right)$, in $O\left(\left(\frac{\gamma(\phi_0)-6L_{\mathrm{cross}}}{L_{S(\mathbf{w},\mathbf{w}_*)}(0)+10L_{\mathrm{cross}}+4\beta}\right)^2\log\left(\frac{1}{\epsilon}\right)\right)$ iterations, we can find the $\epsilon$-close solution of $\mathbf{w}_*$ and the proof is also similar to that of Theorem 3.1.

In practice, we never get a true population gradient but only stochastic gradient $g(\mathbf{w})$ (c.f. Equation (2)). The following theorem shows SGD also recovers the underlying filter.

**Theorem 3.2.** *Let* $\phi_* = \mathrm{argmax}_\phi \gamma(\phi) \geq 6L_{\mathrm{cross}}$. *Denote* $r_0 = \|\mathbf{w}_0 - \mathbf{w}_*\|_2$, $\phi_0 = \arcsin\left(\frac{r_0}{\|\mathbf{w}_*\|_2}\right)$ *and* $\phi_1 = \frac{\phi_*+\phi_0}{2}$. *For* $\epsilon$ *sufficiently small, if* $\eta_t = \Theta\left(\frac{\epsilon^2(\gamma(\phi_1)-6L_{\mathrm{cross}})^2\|\mathbf{w}_*\|_2^2}{B^2}\right)$, *then we have in* $T = O\left(\frac{B^2}{\epsilon^2(\gamma(\phi_1)-6L_{\mathrm{cross}})^2\|\mathbf{w}_*\|_2^2}\log\left(\frac{\|\mathbf{w}_0-\mathbf{w}_*\|}{\epsilon\delta\|\mathbf{w}_*\|_2}\right)\right)$ *iterations, with probability at least* $1-\delta$ *we have* $\|\mathbf{w}_T - \mathbf{w}_*\| \leq \epsilon\|\mathbf{w}_*\|_2$.

Unlike the vanilla gradient descent case, here the convergence rate depends on $\phi_1$ instead of $\phi_0$. This is because of the randomness in SGD and we need a more robust initialization. We choose $\phi_1$ to be the average of $\phi_0$ and $\phi_*$ for the ease of presentation. As will be apparent in the proof we only require $\phi_0$ not very close to $\phi_*$. The proof relies on constructing a martingale and use Azuma-Hoeffding inequality and this idea has been previously used by Ge et al. (2015).

## 3.1 What distribution is easy for SGD to learn a convolutional filter?

Different from One-Layer One-Neuron model, here we also requires the Lipschitz constant for closeness $L_{\mathrm{cross}}$ to be relatively small and $\gamma(\phi_0)$ to be relatively large. A natural question is: What input distributions satisfy this condition?

Here we give an example. We show if (1) patches are close to each other (2) the input distribution has small probability mass around the decision boundary then the assumption in Theorem 3.1 is satisfied. See Figure 1 (b)-(c) for the graphical illustrations.

**Theorem 3.3.** *Denote* $\mathbf{Z}_{avg} = \frac{1}{k}\sum_{i=1}^k \mathbf{Z}_i$. *Suppose all patches have unit norm* [3] *and for all for all* $i$, $\theta(\mathbf{Z}_i, \mathbf{Z}_{avg}) \leq \rho$. *Further assume there exists* $L \geq 0$ *such that for any* $\phi \leq \rho$ *and for all* $\mathbf{Z}_i$

$$\mathbb{P}\left[\theta(\mathbf{Z}_i, \mathbf{w}_*) \in \left[\frac{\pi}{2}-\phi, \frac{\pi}{2}+\phi\right]\right] \leq \mu\phi, \quad \mathbb{P}\left[\theta(\mathbf{Z}_i, \mathbf{w}_*) \in -\left[\frac{\pi}{2}-\phi, -\frac{\pi}{2}+\phi\right]\right] \leq \mu\phi,$$

*then we have*

$$\gamma(\phi_0) \geq \gamma_{avg}(\phi_0) - 4(1-\cos\rho) \text{ and } L_{\mathrm{cross}} \leq 3\mu.$$

*where* $\gamma_{avg}(\phi_0) = \sigma_{\min}\left(\mathbb{E}\left[\mathbf{Z}\mathbf{Z}^\top \mathbb{I}\left\{\mathbf{w}_0^\top \mathbf{Z} \geq 0, \mathbf{w}_*^\top \mathbf{Z} \geq 0\right\}\right]\right)$, *analogue to Definition 2.1.*

Several comments are in sequel. We view $\rho$ as a quantitative measure of the closeness between different patches, i.e., $\rho$ small means they are similar. This lower bound is monotonically decreasing as a function of $\rho$ and note when $\rho = 0$, $\sigma_{\min}\left(\mathbb{E}\left[\mathbf{Z}_{S(\mathbf{w},\mathbf{w}_*)}\mathbf{Z}_{S(\mathbf{w},\mathbf{w}_*)}^\top\right]\right) = \gamma_{avg}(\phi_0)$ which recovers Definition 2.1.

For the upper bond on $L_{\mathrm{cross}}$, $\mu$ represents the upper bound of the probability density around the decision boundary. For example if $\mathbb{P}\left[\theta(\mathbf{Z}_i, \mathbf{w}_*) \in \left[\frac{\pi}{2}-\phi, \frac{\pi}{2}+\phi\right]\right] \propto \phi^2$, then for $\phi$ in a small neighborhood around $\pi/2$, say radius $\epsilon$, we have $\mathbb{P}\left[\theta(\mathbf{Z}_i, \mathbf{w}_*) \in \left[\frac{\pi}{2}-\phi, \frac{\pi}{2}+\phi\right]\right] \lesssim \epsilon\phi$. This assumption is usually satisfied in real world examples like images because the image patches are not usually close to the decision boundary. For example, in computer vision, the local image patches often form clusters and is not evenly distributed over the appearance space. Therefore, if we use linear classifier to separate their cluster centers from the rest of the clusters, near the decision boundary the probability mass should be very low.

## 3.2 The Power of Random Initialization

For one-layer one-neuron model, we need initialization $\|\mathbf{w}_0 - \mathbf{w}_*\|_2 < \|\mathbf{w}_*\|_2$ and for the convolution filter, we need a stronger initialization $\|\mathbf{w}_0 - \mathbf{w}_*\|_2 < \|\mathbf{w}_*\|_2\cos(\phi_*)$. The following theorem

---

[3]This is condition can be relaxed to the norm and the angle of each patch are independent and the norm of each pair is independent of others.

shows with uniformly random initialization we have constant probability to obtain a good initialization. Note with this theorem at hand, we can boost the success probability to arbitrary close to 1 by random restarts. The proof is similar to (Tian, 2017).

**Theorem 3.4.** *If we uniformly sample* $\mathbf{w}_0$ *from a p-dimensional ball with radius* $\alpha\|\mathbf{w}_*\|$ *so that* $\alpha \le \sqrt{\frac{1}{2\pi p}}$, *then with probability at least* $\frac{1}{2} - \sqrt{\frac{\pi p}{2}}\alpha$, *we have* $\|\mathbf{w}_0 - \mathbf{w}_*\|_2 \le \sqrt{1 - \alpha^2}\|\mathbf{w}_*\|$.

To apply this general initialization theorem to our convolution filter case, we can choose $\alpha = \cos\phi_*$. Therefore, with some simple algebra we have the following corollary.

**Corollary 3.1.** *Suppose* $\cos(\phi_*) < \frac{1}{\sqrt{8\pi p}}$, *then if* $\mathbf{w}_0$ *is uniformly sampled from a ball with center* $\mathbf{0}$ *and radius* $\|\mathbf{w}_*\| \cos(\phi_*)$, *we have with probability at least* $\frac{1}{2} - \cos(\phi_*)\sqrt{\frac{\pi p}{2}} > \frac{1}{4}$.

The assumption of this corollary is satisfied if the patches are close to each other as discussed in the previous section.

## 4 EXPERIMENTS

In this section we use simulations to verify our theoretical findings. We first test how the smoothness affect the convergence rate in one-layer one-neuron model described in Section 2 To construct input distribution with different $L(\phi)$, $\gamma(\phi)$ and $\beta$ (c.f. Definition 2.1 and Assumption 2.1), we fix the patch to have unit norm and use a mixture of truncated Gaussian distribution to model on the angle around $\mathbf{w}_*$ and around the $-\mathbf{w}_*$ Specifically, the probability density of $\angle\mathbf{Z}, \mathbf{w}_*$ is sampled from $\frac{1}{2}N(0,\sigma)\mathbb{I}_{[-\pi/2,\pi/2]} + \frac{1}{2}N(-\pi,\sigma)\mathbb{I}_{[-\pi/2,\pi/2]}$. Note by definitions of $L(\phi)$ and $\gamma(\phi)$ if $\sigma \to 0$ the probability mass is centered around $\mathbf{w}_*$, so the distribution is very spiky and $L(\phi)/\gamma(\phi)$ and $\beta$ will be large. On the other hand, if $\sigma \to \infty$, then input distribution is close to the rotation invariant distribution and $L(\phi)/\gamma(\phi)$ and $\beta$ will be small. Figure 3a verifies our prediction where we fix the initialization and step size.

Next we test how the closeness of patches affect the convergence rate in the convolution setting. We first generate a single patch $\widetilde{\mathbf{Z}}$ using the above model with $\sigma = 1$, then generate each unit norm $\mathbf{Z}_i$ whose angle with $\widetilde{\mathbf{Z}}$, $\angle\mathbf{Z}_i, \widetilde{\mathbf{Z}}$ is sampled from $\angle\mathbf{Z}_i, \widetilde{\mathbf{Z}} \sim N(0,\sigma_2)\mathbb{I}_{[-\pi,\pi)}$. Figure 3b shows as variance between patches becomes smaller, we obtain faster convergence rate, which coincides with Theorem 3.1.

We also test whether SGD can learn a filter on real world data. Here we choose MNIST data and generate labels using two filters. One is random filter where each entry is sampled from a standard Gaussian distribution (Figure 4a) and the other is a Gabor filter (Figure 4b). Figure 3a and Figure 3c show convergence rates of SGD with different initializations. Here, better initializations give faster rates, which coincides our theory. Note that here we report the relative loss, logarithm of squared error divided by the square of mean of data points instead of the difference between learned filter and true filter because we found SGD often cannot converge to the exact filter but rather a filter with near zero loss. We believe this is because the data are approximately lying in a low dimensional manifold in which the learned filter and the true filter are equivalent. To justify this conjecture, we try to interpolate the learned filter and the true filter linearly and the result filter has similar low loss (c.f. Figure 5). Lastly, we visualize the true filters and the learned filters in Figure 4 and we can see that the they have similar patterns.

## 5 CONCLUSIONS AND FUTURE WORKS

In this paper we provide the first recovery guarantee of (stochastic) gradient descent algorithm with random initialization for learning a convolution filter when the input distribution is not Gaussian. Our analyses only used the definition of ReLU and some mild structural assumptions on the input distribution. Here we list some future directions.

One possibility is to extend our result to deeper and wider architectures. Even for two-layer fully-connected network, the convergence of (stochastic) gradient descent with random initialization is not known. Existing results either requires sufficiently good initialization (Zhong et al., 2017) or

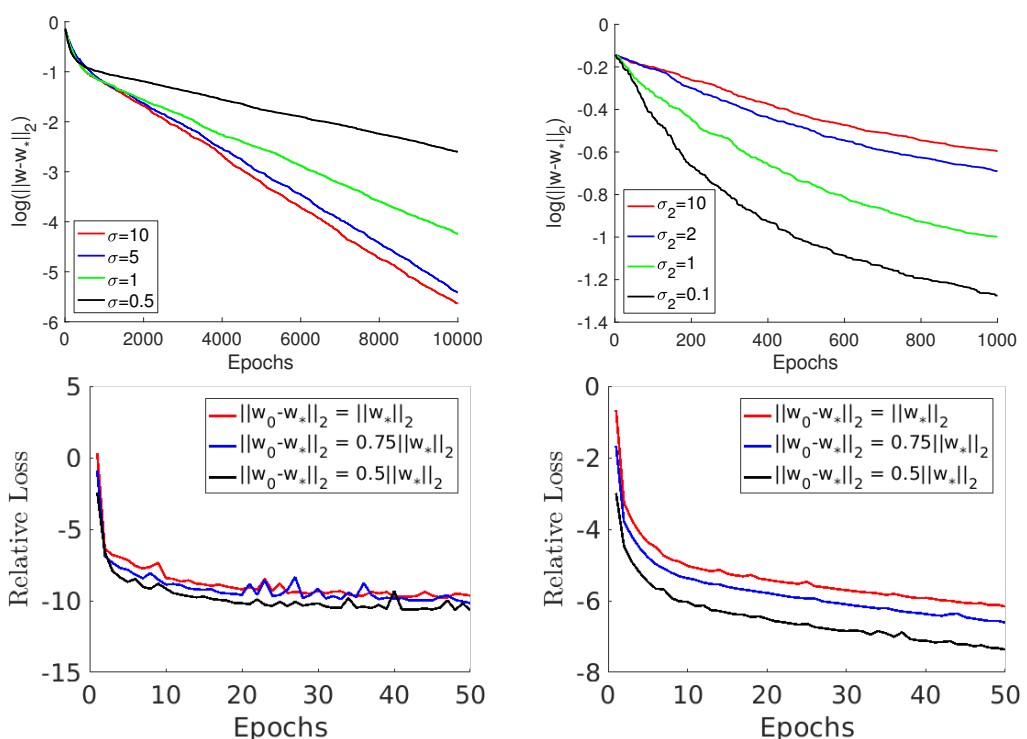

Figure 3: Convergence rates of SGD **(a)** with different smoothness where larger $\sigma$ is smoother; **(b)** with different closeness of patches where smaller $\sigma_2$ is closer; (c) for a learning a random filter with different initialization on MNIST data; **(d)** for a learning a Gabor filter with different initialization on MNIST data.

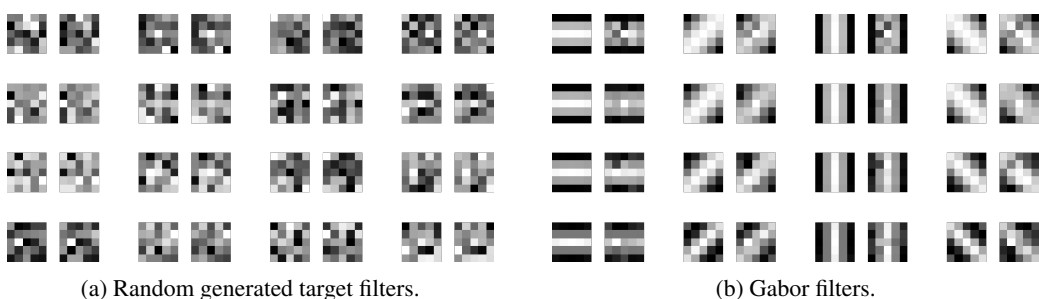

(a) Random generated target filters.       (b) Gabor filters.

Figure 4: Visualization of true and learned filters. For each pair, the left one is the underlying truth and the right is the filter learned by SGD.

relies on special architecture (Li & Yuan, 2017). However, we believe the insights from this paper is helpful to understand the behaviors of gradient-based algorithms in these settings.

Another direction is to consider the agnostic setting, where the label is not equal to the output of a neural network. This will lead to different dynamics of (stochastic) gradient descent and we may need to analyze the robustness of the optimization procedures. This problem is also related to the expressiveness of the neural network (Raghu et al., 2016) where if the underlying function is not equal bot is close to a neural network. We believe our analysis can be extend to this setting.

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

# A    PROOFS AND ADDITIONAL THEOREMS

## A.1    PROOFS OF THE THEOREM IN SECTION 2

**Lemma A.1.**

$$\langle \nabla_{\mathbf{w}}\ell\left(\mathbf{w}\right), \mathbf{w} - \mathbf{w}_* \rangle = \left(\mathbf{w} - \mathbf{w}_*\right)^{\top} \mathbf{A}_{\mathbf{w},\mathbf{w}_*}\left(\mathbf{w} - \mathbf{w}_*\right) + \left(\mathbf{w} - \mathbf{w}_*\right)^{\top} \mathbf{A}_{\mathbf{w},-\mathbf{w}_*}\mathbf{w}. \tag{5}$$

*and both terms are non-negative.*

*Proof.* Since $\mathbf{A}_{\mathbf{w},\mathbf{w}_*} \succeq 0$ and $\mathbf{A}_{\mathbf{w},-\mathbf{w}_*} \succeq 0$ (positive-semidefinite), both the first term and one part of the second term $\mathbf{w}^{\top}\mathbf{A}_{\mathbf{w},-\mathbf{w}_*}\mathbf{w}$ are non-negative. The other part of the second term is

$$-\mathbf{w}_*^{\top}\mathbf{A}_{\mathbf{w},-\mathbf{w}_*}\mathbf{w} = -\mathbb{E}\left[\left(\mathbf{w}_*^{\top}\mathbf{Z}\right)\left(\mathbf{w}^{\top}\mathbf{Z}\right)\mathbb{I}\left\{\mathbf{w}^{\top}\mathbf{Z} \geq 0, \mathbf{w}_*^{\top}\mathbf{Z} \leq 0\right\}\right] \geq 0.$$

$\square$

*Proof of Theorem 2.1.* The assumption on the input distribution ensures when $\theta\left(\mathbf{w}, \mathbf{w}_*\right) \neq \pi$, $\mathbf{A}_{\mathbf{w},\mathbf{w}_*} \succ \mathbf{0}$ and when $\theta\left(\mathbf{w}, \mathbf{w}_*\right) \neq 0$, $\mathbf{A}_{\mathbf{w},-\mathbf{w}_*} \succ \mathbf{0}$. Now when gradient descent converges we have $\nabla_{\mathbf{w}}\ell\left(\mathbf{w}\right) = \mathbf{0}$. We have the following theorem. By assumption, since $\ell\left(\mathbf{w}\right) < \ell\left(\mathbf{0}\right)$ and gradient descent only decreases function value, we will not converge to $\mathbf{w} = \mathbf{0}$. Note that at any critical points, $\langle \nabla_{\mathbf{w}}\ell\left(\mathbf{w}\right), \mathbf{w} - \mathbf{w}_* \rangle = 0$, from Lemma A.1, we have:

$$\left(\mathbf{w} - \mathbf{w}_*\right)^{\top}\mathbf{A}_{\mathbf{w},\mathbf{w}_*}\left(\mathbf{w} - \mathbf{w}_*\right) \quad = 0 \tag{6}$$

$$\left(\mathbf{w} - \mathbf{w}_*\right)^{\top}\mathbf{A}_{\mathbf{w},-\mathbf{w}_*}\mathbf{w} \quad = 0. \tag{7}$$

Suppose we are converging to a critical point $\mathbf{w} \neq \mathbf{w}_*$. There are two cases:

- If $\theta\left(\mathbf{w}, \mathbf{w}_*\right) \neq \pi$, then we have $\left(\mathbf{w} - \mathbf{w}_*\right)^{\top}\mathbf{A}_{\mathbf{w},\mathbf{w}_*}\left(\mathbf{w} - \mathbf{w}_*\right) > 0$, which contradicts with Eqn. 6.

- If $\theta\left(\mathbf{w}, \mathbf{w}_*\right) = \pi$, without loss of generality, let $\mathbf{w} = -\alpha\mathbf{w}_*$ for some $\alpha > 0$. By the assumption we know $\mathbf{A}_{\mathbf{w},-\mathbf{w}_*} \succ \mathbf{0}$. Now the second equation becomes $\left(\mathbf{w} - \mathbf{w}_*\right)^{\top}\mathbf{A}_{\mathbf{w},-\mathbf{w}_*}\mathbf{w} = (1 + \gamma)\mathbf{w}_*\mathbf{A}_{\mathbf{w},-\mathbf{w}_*}\mathbf{w}_* > 0$, which contradicts with Eqn. 7.

Therefore we have $\mathbf{w} = \mathbf{w}_*$.

$\square$

*Proof of Theorem 2.2.* Our proof relies on the following simple but crucial observation: if $\|\mathbf{w} - \mathbf{w}_*\|_2 < \|\mathbf{w}_*\|_2$, then

$$\theta\left(\mathbf{w}, \mathbf{w}_*\right) \leq \arcsin\left(\frac{\|\mathbf{w} - \mathbf{w}_*\|_2}{\|\mathbf{w}_*\|_2}\right).$$

We denote $\theta\left(\mathbf{w}_t, \mathbf{w}_*\right) = \theta_t$ and by the observation we have $\theta_t \leq \phi_t$. Recall the gradient descent dynamics,

$$\mathbf{w}_{t+1} = \mathbf{w}_t - \eta\nabla_{\mathbf{w}_t}\ell(\mathbf{w}_t)$$
$$= \mathbf{w}_t - \eta\left(\mathbb{E}\left[\mathbf{Z}\mathbf{Z}^{\top}\mathbb{I}\left\{\mathbf{w}_t^{\top}\mathbf{Z} \geq 0, \mathbf{w}_*^{\top}\mathbf{Z} \geq 0\right\}\right]\left(\mathbf{w}_t - \mathbf{w}_*\right) - \mathbb{E}\left[\mathbf{w}^{\top}\mathbf{Z} \geq 0, \mathbf{w}_*^{\top}\mathbf{Z} \leq 0\right]\mathbf{w}_t\right).$$

Consider the squared distance to the optimal weight

$$\|\mathbf{w}_{t+1} - \mathbf{w}_*\|_2^2$$
$$= \|\mathbf{w}_t - \mathbf{w}_*\|_2^2$$
$$- \eta\left(\mathbf{w_t} - \mathbf{w}_*\right)^{\top}\left(\mathbb{E}\left[\mathbf{Z}\mathbf{Z}^{\top}\mathbb{I}\left\{\mathbf{w}_t^{\top}\mathbf{Z} \geq 0, \mathbf{w}_*^{\top}\mathbf{Z} \geq 0\right\}\right]\left(\mathbf{w}_t - \mathbf{w}_*\right) - \mathbb{E}\left[\mathbf{w}^{\top}\mathbf{Z} \geq 0, \mathbf{w}_*^{\top}\mathbf{Z} \leq 0\right]\mathbf{w}_t\right)$$
$$+ \eta^2\left\|\mathbb{E}\left[\mathbf{Z}\mathbf{Z}^{\top}\mathbb{I}\left\{\mathbf{w}_t^{\top}\mathbf{Z} \geq 0, \mathbf{w}_*^{\top}\mathbf{Z} \geq 0\right\}\right]\left(\mathbf{w}_t - \mathbf{w}_*\right) - \mathbb{E}\left[\mathbf{w}^{\top}\mathbf{Z} \geq 0, \mathbf{w}_*^{\top}\mathbf{Z} \leq 0\right]\mathbf{w}_t\right\|_2^2.$$

By our analysis in the previous section, the second term is smaller than

$$-\eta\left(\mathbf{w_t} - \mathbf{w}_*\right)^{\top}\mathbb{E}\left[\mathbf{Z}\mathbf{Z}^{\top}\mathbb{I}\left\{\mathbf{w}_t^{\top}\mathbf{Z} \geq 0, \mathbf{w}_*^{\top}\mathbf{Z} \geq 0\right\}\right]\left(\mathbf{w}_t - \mathbf{w}_*\right) \leq -\eta\gamma(\theta_t)\|\mathbf{w}_t - \mathbf{w}_*\|_2^2$$

where we have used our assumption on the angle. For the third term, we expand it as

$$\left\| \mathbb{E}\left[\mathbf{Z}\mathbf{Z}^\top \mathbb{I}\left\{\mathbf{w}_t^\top \mathbf{Z} \geq 0, \mathbf{w}_*^\top \mathbf{Z} \geq 0\right\}\right](\mathbf{w}_t - \mathbf{w}_*) - \mathbb{E}\left[\mathbf{w}^\top \mathbf{Z} \geq 0, \mathbf{w}_*^\top \mathbf{Z} \leq 0\right]\mathbf{w}_t \right\|_2^2$$

$$= \left\| \mathbb{E}\left[\mathbf{Z}\mathbf{Z}^\top \mathbb{I}\left\{\mathbf{w}_t^\top \mathbf{Z} \geq 0, \mathbf{w}_*^\top \mathbf{Z} \geq 0\right\}\right](\mathbf{w}_t - \mathbf{w}_*)\right\|_2^2$$

$$\quad - 2\left(\mathbb{E}\left[\mathbf{Z}\mathbf{Z}^\top \mathbb{I}\left\{\mathbf{w}_t^\top \mathbf{Z} \geq 0, \mathbf{w}_*^\top \mathbf{Z} \geq 0\right\}\right](\mathbf{w}_t - \mathbf{w}_*)\right)^\top \mathbb{E}\left[\mathbf{w}^\top \mathbf{Z} \geq 0, \mathbf{w}_*^\top \mathbf{Z} \leq 0\right]\mathbf{w}_t$$

$$\quad + \left\|\mathbb{E}\left[\mathbf{w}^\top \mathbf{Z} \geq 0, \mathbf{w}_*^\top \mathbf{Z} \leq 0\right]\mathbf{w}_t\right\|_2^2$$

$$\leq L^2(\theta_t)\|\mathbf{w}_t - \mathbf{w}_*\|_2^2 + 2L(\theta_t)\|\mathbf{w}_t - \mathbf{w}_*\|_2 \cdot 2\beta\frac{\|\mathbf{w}_t - \mathbf{w}_*\|}{\|\mathbf{w}_*\|_2} + \left(2\beta\frac{\|\mathbf{w}_t - \mathbf{w}_*\|_2}{\|\mathbf{w}_*\|_2}\right)^2\|\mathbf{w}_t\|_2^2$$

$$\leq L^2(\theta_t)\|\mathbf{w}_t - \mathbf{w}_*\|_2^2 + 2L(\theta_t)\|\mathbf{w}_t - \mathbf{w}_*\| \cdot 2\beta\frac{\|\mathbf{w}_t - \mathbf{w}_*\|}{\|\mathbf{w}_*\|_2} \cdot 2\|\mathbf{w}_*\|_2 + \left(2\beta\frac{\|\mathbf{w}_t - \mathbf{w}_*\|_2}{\|\mathbf{w}_*\|_2}\right)^2 \cdot 4\|\mathbf{w}_*\|_2^2$$

$$\leq \left(L^2(\theta_t) + 8L(\theta_t)\beta + 16\beta^2\right)\|\mathbf{w} - \mathbf{w}_*\|_2^2.$$

Therefore, in summary,

$$\|\mathbf{w}_{t+1} - \mathbf{w}_*\|_2^2 \leq \left(1 - \eta\gamma(\theta_t) + \eta^2\left(L(\theta_t) + 4\beta\right)^2\right)\|\mathbf{w}_t - \mathbf{w}_*\|_2^2$$

$$\leq \left(1 - \frac{\eta\gamma(\theta_t)}{2}\right)\|\mathbf{w}_t - \mathbf{w}_*\|_2^2$$

$$\leq \left(1 - \frac{\eta\gamma(\phi_t)}{2}\right)\|\mathbf{w}_t - \mathbf{w}_*\|_2^2$$

where the first inequality is by our assumption of the step size and second is because $\theta_t \leq \phi_t$ and $\gamma(\cdot)$ is monotonically decreasing. $\qquad\square$

**Theorem A.1** (Rotational Invariant Distribution). *For any unit norm rotational invariant input distribution, we have $\beta = 1$.*

*Proof of Theorem A.1.* Without loss of generality, we only need to focus on the plane spanned by $\mathbf{w}$ and $\mathbf{w}_*$ and suppose $\mathbf{w}_* = (1,0)^\top$. Then

$$\mathbb{E}\left[\mathbf{Z}\mathbf{Z}^\top \mathbb{I}\left\{S(\mathbf{w}, -\mathbf{w}_*)\right\}\right] = \int_{-\pi/2}^{-\pi/2+\phi}\begin{pmatrix}\cos\theta\\\sin\theta\end{pmatrix}(\cos\theta, \sin\theta)\mathrm{d}\theta = \frac{1}{2}\begin{pmatrix}\phi - \sin\phi\cos\phi & -\sin^2\phi\\ -\sin^2\phi & \phi + \sin\phi\cos\phi\end{pmatrix}.$$

It has two eigenvalues

$$\lambda_1(\phi) = \frac{\phi + \sin\phi}{2} \text{ and } \lambda_2(\phi) = \frac{\phi - \sin\phi}{2}.$$

Therefore, $\max_{\mathbf{w},\theta(\mathbf{w},\mathbf{w}_*)\leq\phi}\lambda_{\max}(\mathbf{A}_{\mathbf{w},-\mathbf{w}_*}) = \frac{\phi+\sin\phi}{2} \leq \phi$ for $0 \leq \phi \leq \pi$. $\qquad\square$

**Theorem A.2.** *If $\mathbf{Z} \sim N(0, \mathbf{I})$, then $\beta \leq p$*

*Proof.* Note in previous theorem we can integrate angle and radius separately then multiply them together. For Gaussian distribution, we have $\mathbb{E}\left[\|\mathbf{Z}\|_2^2\right] \leq p$. The result follows. $\qquad\square$

## A.2 PROOFS OF THEOREMS IN SECTION 3

*Proof of Theorem 3.1.* The proof is very similar to Theorem 2.2. Notation-wise, for two events $S_1, S_2$ we use $S_1 S_2$ as a shorthand for $S_1 \cap S_2$ and $S_1 + S_2$ as a shorthand for $S_1 \cup S_2$. Denote $\theta_t = \theta(\mathbf{w}_t, \mathbf{w}_*)$. First note with some routine algebra, we can write the gradient as

$$\nabla_{\mathbf{w}_t}\ell(\mathbf{w}_t)$$

$$= \mathbb{E}\left[\sum_{(i,j)=(1,1)}^{(d,d)} \mathbf{Z}_i \mathbf{Z}_j^\top \mathbb{I}\left\{S(\mathbf{w}, \mathbf{w}_*)_i S(\mathbf{w}, \mathbf{w}_*)_j\right\}\right](\mathbf{w} - \mathbf{w}_*)$$

$$+ \mathbb{E}\left[\sum_{(i,j)=(1,1)}^{(d,d)} \mathbf{Z}_i \mathbf{Z}_j^\top \mathbb{I}\left\{S(\mathbf{w},\mathbf{w}_*)_i S(\mathbf{w},-\mathbf{w}_*)_j + S(\mathbf{w},-\mathbf{w}_*)_i S(\mathbf{w},\mathbf{w}_*)_j\right\}\right]\mathbf{w}$$

$$+ \mathbb{E}\left[\sum_{(i,j)=(1,1)}^{(d,d)} \mathbf{Z}_i \mathbf{Z}_j^\top \mathbb{I}\left\{S(\mathbf{w},-\mathbf{w}_*)_i S(\mathbf{w},-\mathbf{w}_*)_j\right\}\right]\mathbf{w}$$

$$- \mathbb{E}\left[\sum_{(i,j)=(1,1)}^{(d,d)} \mathbf{Z}_i \mathbf{Z}_j^\top \mathbb{I}\left\{S(\mathbf{w},\mathbf{w}_*)_i S(-\mathbf{w},\mathbf{w}_*)_j + S(\mathbf{w},-\mathbf{w}_*)_i S(\mathbf{w},\mathbf{w}_*)_j + S(\mathbf{w},-\mathbf{w}_*)_i S(-\mathbf{w},\mathbf{w}_*)_j\right\}\right]\mathbf{w}_*$$

We first examine the inner product between the gradient and $\mathbf{w}-\mathbf{w}_*$.

$$\langle \nabla_{\mathbf{w}_t}\ell(\mathbf{w}), \mathbf{w}-\mathbf{w}_*\rangle$$

$$= (\mathbf{w}-\mathbf{w}_*)^\top \mathbb{E}\left[\sum_{(i,j)=(1,1)}^{(d,d)} \mathbf{Z}_i \mathbf{Z}_j \mathbb{I}\left\{S(\mathbf{w},\mathbf{w}_*)_i S(\mathbf{w},\mathbf{w}_*)_j\right\}\right](\mathbf{w}-\mathbf{w}_*)$$

$$+ (\mathbf{w}-\mathbf{w}_*)^\top \mathbb{E}\left[\sum_{(i,j)=(1,1)}^{(d,d)} \mathbf{Z}_i \mathbf{Z}_j \mathbb{I}\left\{S(\mathbf{w},\mathbf{w}_*)_i S(\mathbf{w},-\mathbf{w}_*)_j + S(\mathbf{w},-\mathbf{w}_*)_i S(\mathbf{w},\mathbf{w}_*)_j + S(\mathbf{w},-\mathbf{w}_*)_i S(\mathbf{w},-\mathbf{w}_*)_j\right\}\right]\mathbf{w}$$

$$- (\mathbf{w}-\mathbf{w}_*)^\top \mathbb{E}\left[\sum_{(i,j)=(1,1)}^{(d,d)} \mathbf{Z}_i \mathbf{Z}_j \mathbb{I}\left\{S(\mathbf{w},\mathbf{w}_*)_i S(-\mathbf{w},\mathbf{w}_*)_j + S(\mathbf{w},-\mathbf{w}_*)_i S(\mathbf{w},\mathbf{w}_*)_j + S(\mathbf{w},-\mathbf{w}_*)_i S(-\mathbf{w},\mathbf{w}_*)_j\right\}\right]\mathbf{w}_*$$

$$\geq (\mathbf{w}-\mathbf{w}_*)^\top \mathbb{E}\left[\sum_{(i,j)=(1,1)}^{(d,d)} \mathbf{Z}_i \mathbf{Z}_j \mathbb{I}\left\{S(\mathbf{w},\mathbf{w}_*)_i S(\mathbf{w},\mathbf{w}_*)_j\right\}\right](\mathbf{w}-\mathbf{w}_*)$$

$$+ (\mathbf{w}-\mathbf{w}_*)^\top \mathbb{E}\left[\sum_{(i,j)=(1,1)}^{(d,d)} \mathbf{Z}_i \mathbf{Z}_j \mathbb{I}\left\{S(\mathbf{w},\mathbf{w}_*)_i S(\mathbf{w},-\mathbf{w}_*)_j\right\}\right]\mathbf{w}$$

$$- (\mathbf{w}-\mathbf{w}_*)^\top \mathbb{E}\left[\sum_{(i,j)=(1,1)}^{(d,d)} \mathbf{Z}_i \mathbf{Z}_j \mathbb{I}\left\{S(\mathbf{w},\mathbf{w}_*)_i S(-\mathbf{w},\mathbf{w}_*)_j + S(\mathbf{w},-\mathbf{w}_*)_i S(\mathbf{w},\mathbf{w}_*)_j + S(\mathbf{w},-\mathbf{w}_*)_i S(-\mathbf{w},\mathbf{w}_*)_j\right\}\right]\mathbf{w}_*$$

$$\geq \gamma(\theta_t)\|\mathbf{w}-\mathbf{w}_*\|_2^2$$

$$- \|\mathbf{w}-\mathbf{w}_*\|_2 \|\mathbf{w}\|_2 \left\|\mathbb{E}\left[\sum_{(i,j)=(1,1)}^{(d,d)} \mathbf{Z}_i \mathbf{Z}_j \mathbb{I}\left\{S(\mathbf{w},\mathbf{w}_*)_i S(\mathbf{w},-\mathbf{w}_*)_j\right\}\right]\right\|_{op}$$

$$- \|\mathbf{w}-\mathbf{w}_*\|_2 \|\mathbf{w}_*\|_2 \left(\left\|\mathbb{E}\left[\sum_{(i,j)=(1,1)}^{(d,d)} \mathbf{Z}_i \mathbf{Z}_j \mathbb{I}\left\{S(\mathbf{w},\mathbf{w}_*)_i S(-\mathbf{w},\mathbf{w}_*)_j\right\}\right]\right\|_{op}\right.$$

$$\left. + \left\|\mathbb{E}\left[\sum_{(i,j)=(1,1)}^{(d,d)} \mathbf{Z}_i \mathbf{Z}_j \mathbb{I}\left\{S(\mathbf{w},-\mathbf{w}_*)_i S(\mathbf{w},\mathbf{w}_*)_j\right\}\right]\right\|_{op} + \left\|\mathbb{E}\left[\sum_{(i,j)=(1,1)}^{(d,d)} \mathbf{Z}_i \mathbf{Z}_j \mathbb{I}\left\{S(\mathbf{w},-\mathbf{w}_*)_i S(-\mathbf{w},\mathbf{w}_*)_j\right\}\right]\right\|_{op}\right)$$

$$\geq \gamma(\theta_t)\|\mathbf{w}-\mathbf{w}_*\|_2^2$$

$$- 2\|\mathbf{w}-\mathbf{w}_*\|_2 \|\mathbf{w}_*\|_2 \left\|\mathbb{E}\left[\sum_{(i,j)=(1,1)}^{(d,d)} \mathbf{Z}_i \mathbf{Z}_j \mathbb{I}\left\{S(\mathbf{w},\mathbf{w}_*)_i S(\mathbf{w},-\mathbf{w}_*)_j\right\}\right]\right\|_{op}$$

$$- \|\mathbf{w}-\mathbf{w}_*\|_2 \|\mathbf{w}_*\|_2 \left(\left\|\mathbb{E}\left[\sum_{(i,j)=(1,1)}^{(d,d)} \mathbf{Z}_i \mathbf{Z}_j \mathbb{I}\left\{S(\mathbf{w},\mathbf{w}_*)_i S(-\mathbf{w},\mathbf{w}_*)_j\right\}\right]\right\|_{op}\right.$$

$$+ \left\| \mathbb{E}\left[ \sum_{(i,j)=(1,1)}^{(d,d)} \mathbf{Z}_i\mathbf{Z}_j \mathbb{I}\left\{ S(\mathbf{w}, -\mathbf{w}_*)_i S(\mathbf{w}, \mathbf{w}_*)_j \right\}\right] \right\|_{op} + \left\| \mathbb{E}\left[ \sum_{(i,j)=(1,1)}^{(d,d)} \mathbf{Z}_i\mathbf{Z}_j \mathbb{I}\left\{ S(\mathbf{w}, -\mathbf{w}_*)_i S(-\mathbf{w}, \mathbf{w}_*)_j \right\}\right] \right\|_{op} \right)$$

$$\geq \gamma\left(\theta_t\right) \|\mathbf{w}_t - \mathbf{w}_*\|_2^2 - 3L_{\mathrm{cross}}\phi_t \|\mathbf{w}_*\|_2 \|\mathbf{w}_t - \mathbf{w}_*\|_2$$

$$\geq \gamma\left(\theta_t\right) \|\mathbf{w}_t - \mathbf{w}_*\|_2^2 - 6L_{\mathrm{cross}}\frac{\|\mathbf{w}_t - \mathbf{w}_*\|_2}{\|\mathbf{w}_*\|_2} \cdot \|\mathbf{w}_*\|_2 \|\mathbf{w}_t - \mathbf{w}_*\|_2$$

$$\geq \left(\gamma\left(\theta_t\right) - 6L_{\mathrm{cross}}\right) \|\mathbf{w}_t - \mathbf{w}_*\|_2^2$$

where the first inequality we used the definitions of the regions; the second inequality we used the definition of operator norm; the third inequality we used the fact $\|\mathbf{w}_t - \mathbf{w}_*\|_2 \leq \|\mathbf{w}_*\|_2$; the fourth inequality we used the definition of $L_{\mathrm{cross}}$ and the fifth inequality we used $\phi \leq 2\sin\phi$ for any $0 \leq \phi \leq \pi/2$. Next we can upper bound the norm of the gradient using similar argument

$$\|\nabla_{\mathbf{w}_t}\ell(\mathbf{w}_t)\|_2 \leq L\left(\theta_t\right) \|\mathbf{w}_t - \mathbf{w}_*\|_2 + 10L_{\mathrm{cross}} \|\mathbf{w}_t - \mathbf{w}_*\| + 2\beta \|\mathbf{w}_t - \mathbf{w}_*\|_2$$
$$= (L(\theta_t) + 10L_{\mathrm{cross}} + 4\beta) \|\mathbf{w}_t - \mathbf{w}_*\|_2.$$

Therefore, using the dynamics of gradient descent, putting the above two bounds together, we have

$$\|\mathbf{w}_{t+1} - \mathbf{w}_*\|_2^2 \leq \left(1 - \eta\left(\gamma(\theta_t) - 6L_{\mathrm{cross}}\right) + \eta^2(L(\theta_t) + 10L_{\mathrm{cross}} + 4\beta)^2\right) \|\mathbf{w}_t - \mathbf{w}_*\|_2^2$$
$$\leq \left(1 - \frac{\eta(\gamma(\theta_t) - 6L_{\mathrm{cross}})}{2}\right) \|\mathbf{w}_t - \mathbf{w}_*\|_2^2$$
$$\leq \left(1 - \frac{\eta(\gamma(\phi_t) - 6L_{\mathrm{cross}})}{2}\right) \|\mathbf{w}_t - \mathbf{w}_*\|_2^2$$

where the last step we have used our choice of $\eta_t$ and $\theta_t \leq \phi_t$. $\qquad \square$

The proof of Theorem 3.2 consists of two parts. First we show if $\eta$ is chosen properly and $T$ is not to big, then for all $1 \leq t \leq T$, with high probability the iterates stat in a neighborhood of $\mathbf{w}_*$. Next, conditioning on this, we derive the rate.

**Lemma A.2.** *Denote $r_0 = \|\mathbf{w}_0 - \mathbf{w}_*\|_2 < \|\mathbf{w}_*\|_2 \sin\phi_*$. Given $0 < r_1 < \|\mathbf{w}_*\|_2 \sin\phi_*$, number of iterations $T \in \mathbb{Z}_{++}$ and failure probability $\delta$, denote $\phi_1 = \arcsin\left(\frac{r_1}{\|\mathbf{w}_*\|_2}\right)$ then if the step size satisfies*

$$0 < 1 - \eta\gamma(\phi_1) + \eta^2(L(0) + 10L_{\mathrm{cross}} + 4\beta)^2 < 1$$

$$\frac{\left(r_1^2 - r_0^2\right)^2}{T\left(1 + 2\eta\alpha T\right)\left(2\eta B\left(L(0) + 10L_{\mathrm{cross}} + 4\beta\right)r_1 + \eta^2 B^2\right)^2} \geq \log\left(\frac{T}{\delta}\right)$$

*with $\alpha = \gamma\left(\phi_1\right) - \eta\left(L(0) + 10L_{\mathrm{cross}} + 4\beta\right)$. Then with probability at least $1 - \delta$, for all $t = 1, \ldots, T$, we have*

$$\|\mathbf{w}_t - \mathbf{w}_*\| \leq r_1.$$

*Proof of Lemma A.2.* Let $g(\mathbf{w}_t) = \mathbb{E}\left[\nabla_{\mathbf{w}_t}\ell\left(\mathbf{w}_t\right)\right] + \xi_t$. We denote $\mathcal{F}_t = \sigma\left\{\xi_1, \ldots, \xi_t\right\}$, the sigma-algebra generated by $\xi_1, \ldots, \xi_t$ and define the event

$$\mathcal{C}_t = \left\{\forall\tau \leq t, \|\mathbf{w}_\tau - \mathbf{w}_*\| \leq r_1\right\}.$$

Consider

$$\mathbb{E}\left[\|\mathbf{w}_{t+1} - \mathbf{w}_*\|_2^2 \mathbb{I}_{\mathcal{C}_t}|\mathcal{F}_t\right]$$
$$= \mathbb{E}\left[\|\mathbf{w}_t - \eta\nabla_{\mathbf{w}_t}\ell(\mathbf{w}_t) - \mathbf{w}_* - \eta\xi_t\|_2^2 \mathbb{I}_{\mathcal{C}_t}|\mathcal{F}_t\right]$$
$$\leq \left(\left(1 - \eta\gamma(\phi_1) + \eta^2\left(L(0) + 10L_{\mathrm{cross}} + 4\beta\right)^2\right) \|\mathbf{w}_t - \mathbf{w}_*\|_2^2 + \eta^2 B^2\right) \mathbb{I}_{\mathcal{C}_t}$$

where the inequality follows by our analysis of gradient descent together with definition of $\mathcal{C}_t$ and $\mathbb{E}\left[\xi_t|\mathcal{F}_t\right] = 0$. Define

$$G_t = (1 - \eta\alpha)^{-t}\left(\|\mathbf{w}_t - \mathbf{w}_*\|_2^2 - \frac{\eta B^2}{\alpha}\right).$$

By our analysis above, we have

$$\mathbb{E}\left[G_{t+1}\mathbb{I}_{\mathcal{C}_t}|\mathcal{F}_t\right] \leq G_t\mathbb{I}_{\mathcal{C}_t} \leq G_t\mathbb{I}_{\mathcal{C}_{t-1}}$$

where the last inequality is because $\mathcal{C}_t$ is a subset of $\mathcal{C}_{t-1}$. Therefore, $G_t\mathbb{I}_{\mathcal{C}_{t-1}}$ is a super-martingale and we may apply Azuma-Hoeffding inequality. Before that, we need to bound the difference between $G_t\mathbb{I}_{\mathcal{C}_t}$ and its expectation. Note

$$\begin{aligned}
\left|G_t\mathbb{I}_{\mathcal{C}_{t-1}} - \mathbb{E}\left[G_t\mathbb{I}_{\mathcal{C}_{t-1}}\right]|\mathcal{F}_{t-1}\right| &= (1-\eta\alpha)^{-t}\left|\|\mathbf{w}_t - \mathbf{w}_*\|_2^2 - \mathbb{E}\left[\|\mathbf{w}_t - \mathbf{w}_*\|_2^2\right]|\mathcal{F}_{t-1}\right|\mathbb{I}_{\mathcal{C}_{t-1}} \\
&= (1-\eta\alpha)^{-t}\left|2\eta\langle\xi_t, \mathbf{w}_t - \eta\nabla_{\mathbf{w}_t}\ell(\mathbf{w}_t) - \mathbf{w}_* - \eta^2\mathbb{E}\left[\|\xi_t\|_2^2|\mathcal{F}_{t-1}\right]\right|\mathbb{I}_{\mathcal{C}_{t-1}} \\
&\leq (1-\eta\alpha)^{-t}\left(2\eta B\left(L(0) + 10L_{\text{cross}} + 4\beta\right)\|\mathbf{w}_t - \mathbf{w}_*\|_2 + \eta^2 B^2\right)\mathbb{I}_{\mathcal{C}_{t-1}} \\
&\leq (1-\eta\alpha)^{-t}\left(2\eta B\left(L(0) + 10L_{\text{cross}} + 4\beta\right)r_1 + \eta^2 B^2\right) \\
&\triangleq d_t.
\end{aligned}$$

Therefore for all $t \leq T$

$$\begin{aligned}
c_t^2 &\triangleq \sum_{\tau=1}^{t} d_\tau^2 \\
&= \sum_{\tau=1}^{t}(1-\eta\alpha)^{-2t}\left(2\eta B\left(L(0) + 10L_{\text{cross}} + 4\beta\right)r_1 + \eta^2 B^2\right)^2 \\
&\leq t(1-\eta\alpha)^{-2t}\left(2\eta B\left(L(0) + 10L_{\text{cross}} + 4\beta\right)r_1 + \eta^2 B^2\right)^2 \\
&\leq T(1+2\eta\alpha T)\left(2\eta B\left(L(0) + 10L_{\text{cross}} + 4\beta\right)r_1 + \eta^2 B^2\right)^2
\end{aligned}$$

where the first inequality we used $1 - \eta\alpha < 1$, the second we used $t \leq T$ and the third we used our assumption on $\eta$. Let us bound at $(t+1)$-th step, the iterate goes out of the region,

$$\begin{aligned}
\mathbb{P}\left[\mathcal{C}_t \cap \{\|\mathbf{w}_{t+1} - \mathbf{w}_*\|_2 > r_1\}\right] &= \mathbb{P}\left[\mathcal{C}_t \cap \left\{\|\mathbf{w}_{t+1} - \mathbf{w}_*\|_2^2 > r_1^2\right\}\right] \\
&= \mathbb{P}\left[\mathcal{C}_t \cap \left\{\|\mathbf{w}_{t+1} - \mathbf{w}_*\|_2^2 > r_0^2 + (r_1^2 - r_0^2)\right\}\right] \\
&= \mathbb{P}\left[\mathcal{C}_t \cap \left\{G_{t+1}(1-\eta\alpha)^t + \frac{\eta B^2}{\alpha} \geq G_0 + \frac{\eta B^2}{\alpha} + r_1^2 - r_0^2\right\}\right] \\
&\leq \mathbb{P}\left[\mathcal{C}_t \cap \left\{G_{t+1} - G_0 \geq r_1^2 - r_0^2\right\}\right] \\
&\leq \exp\left\{-\frac{(r_1^2 - r_0^2)^2}{2c_t^2}\right\} \\
&\leq \frac{\delta}{T}
\end{aligned}$$

where the second inequality we used Azuma-Hoeffding inequality, the last one we used our assumption of $\eta$. Therefore for all $0 \leq t \leq T$, we have with probability at least $1 - \delta$, $\mathcal{C}_t$ happens. $\qquad\square$

Now we can derive the rate.

**Lemma A.3.** *Denote $r_0 = \|\mathbf{w}_0 - \mathbf{w}_*\|_2 < \|\mathbf{w}_*\|_2\sin\phi_*$. Given $0 < r_1 < \|\mathbf{w}_*\|_2\sin\phi_*$, number of iterations $T \in \mathbb{Z}_{++}$ and failure probability $\delta$, denote $\phi_1 = \arcsin\left(\frac{r_1}{\|\mathbf{w}_*\|_2}\right)$ then if the step size satisfies*

$$0 < 1 - \eta\gamma(\phi_1) + \eta^2(L(0) + 10L_{\text{cross}} + 4\beta)^2 < 1$$

$$\frac{(r_1^2 - r_0^2)^2}{T(1 + 2\eta\alpha T)\left(2\eta B\left(L(0) + 10L_{\text{cross}} + 4\beta\right)r_1 + \eta^2 B^2\right)^2} \geq \log\left(\frac{T}{\delta}\right)$$

$$\eta T\left(\gamma(\phi_1) - \eta\left(L(0) + 10L_{\text{cross}} + 4\beta\right)^2\right) \geq \log\left(\frac{r_0^2}{\epsilon^2\|\mathbf{w}_*\|_2^2\delta}\right)$$

$$\epsilon^2 \left( \gamma(\phi_1) - \eta \left( L(0) + 10L_{\mathrm{cross}} + 4\beta \right)^2 \right) \|\mathbf{w}_*\|_2^2 \geq \eta B^2$$

with $\alpha = \gamma(\phi_1) - \eta\left(L(0) + 10L_{\mathrm{cross}} + 4\beta\right)$, then we have with probability $1 - 2\delta$,

$$\|\mathbf{w}_t - \mathbf{w}_*\|_2 \leq 2\epsilon \|\mathbf{w}_*\|_2 .$$

*Proof of Lemma A.3.* We use the same notations in the proof of Lemma A.2. By the analysis of Lemma A.2, we know

$$\mathbb{E}\left[ \|\mathbf{w}_{t+1} - \mathbf{w}_*\|_2^2 \, \mathbb{I}_{\mathcal{C}_t} | \mathcal{F}_t \right] \leq \left( (1 - \eta\alpha) \|\mathbf{w}_t - \mathbf{w}_*\|_2^2 + \eta^2 B^2 \right) \mathbb{I}_{\mathcal{C}_t}.$$

Therefore we have

$$\mathbb{E}\left[ \|\mathbf{w}_t - \mathbf{w}_*\|_2^2 \, \mathbb{I}_{\mathcal{C}_t} - \frac{\eta B^2}{\alpha} \right] \leq (1 - \eta\alpha)^t \left( \|\mathbf{w}_0 - \mathbf{w}_*\|_2^2 - \frac{\eta B}{\alpha} \right).$$

Now we can bound the failure probability

$$
\begin{aligned}
\mathbb{P}\left[ \|\mathbf{w}_T - \mathbf{w}_*\|_2 \geq 2\epsilon \|\mathbf{w}_*\|_2 \right] \leq & \mathbb{P}\left[ \|\mathbf{w}_T - \mathbf{w}_*\|_2^2 - \frac{\eta B^2}{\alpha} \geq \epsilon^2 \|\mathbf{w}_*\|_2^2 \right] \\
\leq & \mathbb{P}\left[ \left\{ \|\mathbf{w}_T - \mathbf{w}_*\|_2^2 \, \mathbb{I}_{\mathcal{C}_t} - \frac{\eta B^2}{\alpha} \geq \epsilon^2 \|\mathbf{w}\|_2^2 \right\} \cup \mathcal{C}_t^c \right] \\
\leq & \mathbb{P}\left[ \left\{ \|\mathbf{w}_T - \mathbf{w}_*\|_2^2 \, \mathbb{I}_{\mathcal{C}_t} - \frac{\eta B^2}{\alpha} \geq \epsilon^2 \|\mathbf{w}\|_2^2 \right\} \right] + \delta \\
\leq & \frac{\mathbb{E}\left[ \|\mathbf{w}_T - \mathbf{w}_*\|_2^2 \, \mathbb{I}_{\mathcal{C}_t} - \frac{\eta B^2}{\alpha} \right]}{\epsilon^2 \|\mathbf{w}_*\|_2^2} + \delta \\
\leq & \frac{(1 - \eta\alpha)^t \left( \|\mathbf{w}_0 - \mathbf{w}_*\|_2^2 - \frac{\eta B}{\alpha} \right)}{\epsilon^2 \|\mathbf{w}_*\|_2^2} + \delta \\
\leq & 2\delta.
\end{aligned}
$$

The first inequality we used the last assumption. The second inequality we used the probability of an event is upper bound by any superset of this event. The third one we used Lemma A.2 and the union bound. The fourth one we used Markov's inequality. $\qquad\square$

Now we can specify the $T$ and $\eta$ and derive the convergence rate of SGD for learning a convolution filter.

*Proof of Theorem 3.2.* With the choice of $\eta$ and $T$, it is straightforward to check they satisfies conditions in Lemma A.3. $\qquad\square$

*Proof of Theorem 3.3.* We first prove the lower bound of $\gamma(\phi_0)$.

$$
\begin{aligned}
& \mathbb{E}\left[ \left( \sum_{i=1}^{k} \mathbf{Z}_i \mathbb{I}\left\{ S(\mathbf{w}, \mathbf{w}_*)_i \right\} \right) \left( \sum_{i=1}^{k} \mathbf{Z}_i \mathbb{I}\left\{ S(\mathbf{w}, \mathbf{w}_*)_i \right\} \right)^{\top} \right] \\
=& \mathbb{E}\left[ k\mathbf{Z} + \sum_{i=1}^{k} \left( \mathbf{Z}_i \mathbb{I}\left\{ S(\mathbf{w}, \mathbf{w}_*)_i \right\} - \mathbf{Z} \right) \left( k\mathbf{Z} + \sum_{i=1}^{k} \left( \mathbf{Z}_i \mathbb{I}\left\{ S(\mathbf{w}, \mathbf{w}_*)_i \right\} - \mathbf{Z} \right) \right)^{\top} \right] \\
=& k^2 \mathbb{E}\left[ \mathbf{Z}\mathbf{Z}^{\top} \right] + k \mathbb{E}\left[ \mathbf{Z} \left( \sum_{i=1}^{k} \left( \mathbf{Z}_i \mathbb{I}\left\{ S(\mathbf{w}, \mathbf{w}_*)_i \right\} - \mathbf{Z} \right) \right)^{\top} \right] \\
& + k \mathbb{E}\left[ \left( \sum_{i=1}^{k} \left( \mathbf{Z}_i \mathbb{I}\left\{ S(\mathbf{w}, \mathbf{w}_*)_i \right\} - \mathbf{Z} \right) \right) \mathbf{Z}^{\top} \right]
\end{aligned}
$$

$$+ \mathbb{E}\left[\left(\sum_{i=1}^{k}\left(\mathbf{Z}_i\mathbb{I}\left\{S(\mathbf{w},\mathbf{w}_*)_i\right\} - \mathbf{Z}_1\mathbb{I}\left\{S(\mathbf{w},\mathbf{w}_*)_1\right\}\right)\right)\left(\sum_{i=1}^{k}\left(\mathbf{Z}_i\mathbb{I}\left\{S(\mathbf{w},\mathbf{w}_*)_i\right\} - \mathbf{Z}_1\mathbb{I}\left\{S(\mathbf{w},\mathbf{w}_*)_1\right\}\right)\right)^{\top}\right]$$

$$\succeq k^2\mathbb{E}\left[\mathbf{Z}\mathbf{Z}^{\top}\right] + k\mathbb{E}\left[\mathbf{Z}\left(\sum_{i=1}^{k}\left(\mathbf{Z}_i\mathbb{I}\left\{S(\mathbf{w},\mathbf{w}_*)_i\right\} - \mathbf{Z}\right)\right)^{\top}\right]$$

$$+ k\mathbb{E}\left[\left(\sum_{i=1}^{k}\left(\mathbf{Z}_i\mathbb{I}\left\{S(\mathbf{w},\mathbf{w}_*)_i\right\} - \mathbf{Z}\right)\right)\mathbf{Z}^{\top}\right]$$

Note because $\mathbf{Z}_i$s have unit norm and by law of cosines $\left\|\mathbf{Z}\left(\mathbf{Z}_i\mathbb{I}\left\{S(\mathbf{w},\mathbf{w}_*)_i\right\} - \mathbf{Z}\right)\right\|_{op} \leq 2(1 - \cos\rho)$. Therefore,

$$\sigma_{\min}\left(\mathbb{E}\left[\left(\sum_{i=1}^{d}\mathbf{Z}_i\mathbb{I}\left\{S(\mathbf{w},\mathbf{w}_*)_i\right\}\right)\left(\sum_{i=1}^{d}\mathbf{Z}_i\mathbb{I}\left\{S(\mathbf{w},\mathbf{w}_*)_i\right\}\right)^{\top}\right]\right) \geq k^2(\gamma_1(\phi_0) - 4(1 - \cos\rho)).$$

Now we prove the upper bound of $L_{\text{cross}}$. Notice that

$$\left\|\mathbb{E}\left[\mathbf{Z}_i\mathbf{Z}_j^{\top}\mathbb{I}\left\{S(\mathbf{w},\mathbf{w}_*)_i S(\mathbf{w},-\mathbf{w}_*)_j\right\}\right]\right\|_2 \leq \mathbb{E}\left[\|\mathbf{Z}_i\|_2\|\mathbf{Z}_j\|_2\mathbb{I}\left\{S(\mathbf{w},\mathbf{w}_*)_i S(\mathbf{w},-\mathbf{w}_*)_j\right\}\right]$$

$$= \int_{S(\mathbf{w},-\mathbf{w}_*)_j}\left(\int_{S(\mathbf{w},\mathbf{w}_*)_i} d\mathbb{P}\left(\mathbf{Z}_i|\mathbf{Z}_j\right)\right)d\mathbb{P}\left(\theta_j\right).$$

If $\phi \leq \psi$, then by our assumption, we have

$$\int_{S(\mathbf{w},-\mathbf{w}_*)_j}\left(\int_{S(\mathbf{w},\mathbf{w}_*)_i} d\mathbb{P}\left(\mathbf{Z}_i|\mathbf{Z}_j\right)\right)d\mathbb{P}\left(\theta_j\right) \leq \int_{S(\mathbf{w},-\mathbf{w}_*)_j} d\mathbb{P}\left(\mathbf{Z}_j\right) \leq L\phi.$$

On the other hand, if $\phi \geq \gamma$, let $\theta_j$ be the angle between $\mathbf{w}_*$ and $\mathbf{Z}_j$, we have

$$\int_{S(\mathbf{w},-\mathbf{w}_*)_j}\left(\int_{S(\mathbf{w},\mathbf{w}_*)_i} d\mathbb{P}\left(\mathbf{Z}_i|\mathbf{Z}_j\right)\right)d\mathbb{P}\left(\theta_j\right) \leq \int_{\frac{\pi}{2}}^{\frac{\pi}{2}+\gamma}\left(\int_{S(\mathbf{w},\mathbf{w}_*)_i} d\mathbb{P}\left(\mathbf{Z}_i|\mathbf{Z}_j\right)\right)d\mathbb{P}\left(\theta_j\right)$$

$$\leq L\gamma$$
$$\leq L\phi.$$

Therefore, $\sigma_{\max}\left(\mathbb{E}\left[\mathbf{Z}_{S(\mathbf{w},\mathbf{w}_*)}\mathbf{Z}_{S(\mathbf{w},-\mathbf{w}_*)}^{\top}\right]\right) \leq L\phi$. Using similar arguments we can show $\sigma_{\max}\left(\mathbb{E}\left[\mathbf{Z}_{S(\mathbf{w},\mathbf{w}_*)}\mathbf{Z}_{S(-\mathbf{w},\mathbf{w}_*)}\right]\right) \leq L\phi$ and $\sigma_{\max}\left(\mathbb{E}\left[\mathbf{Z}_{S(\mathbf{w},-\mathbf{w}_*)}\mathbf{Z}_{S(-\mathbf{w},\mathbf{w}_*)}\right]\right) \leq L\phi$. $\qquad\square$

*Proof of Theorem 3.4.* We use the same argument by Tian (2017). Let $r_{init}$ be the initialization radius. The failure probability is lower bounded

$$\frac{1}{2}\left(r_{init}\right) - \frac{\left(\frac{r_{init}^2}{2\|\mathbf{w}_*\|_2} + \frac{\|\mathbf{w}_*\|_2\cos(\phi_*)}{2}\right)\delta V_{k-1}\left(r_{init}\right)}{V_k\left(r_{init}\right)}.$$

Therefore, $r_{init} = \cos(\phi_*)\|\mathbf{w}_*\|_2$ maximizes this lower bound. Plugging this optimizer in and using formula for the volume of the Euclidean ball, the failure probability is lower bounded by

$$\frac{1}{2} - \cos(\phi_*)\frac{\pi\Gamma(p/2 + 1)}{\Gamma(p/2 + 1/2)} \geq \frac{1}{2} - \cos(\phi_*)\sqrt{\frac{\pi p}{2}}$$

where we used Gautschi's inequality for the last step. $\qquad\square$

# B  ADDITIONAL EXPERIMENTAL RESULTS

Figure 5 show the loss of linear interpolation between the learned filter $w$ and ground truth filter $w_*$. Our interpolation has the form $w_{inter} = \alpha w + (1 - \alpha)w_*$ where $\alpha \in [0, 1]$ is the interpolation ratio. Note that for all interpolation ratios, the loss remains very low.

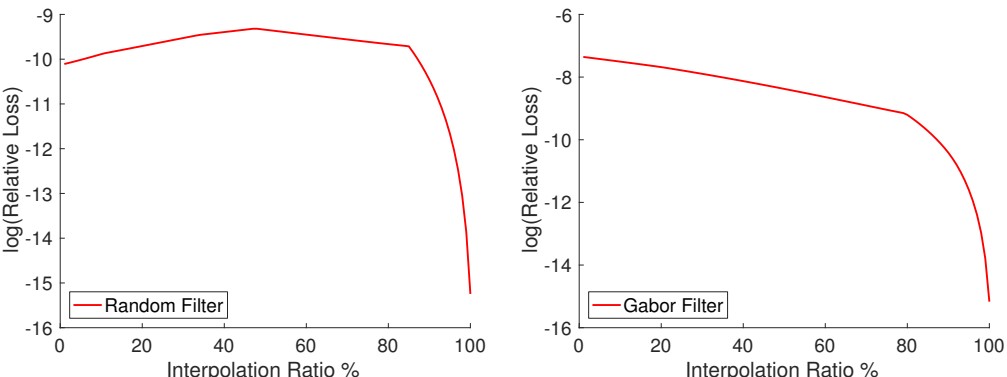

Figure 5: Loss of linear interpolation between learned filter and the true filter.

