# OpenReview forum: "When is a Convolutional Filter Easy to Learn?"
_ICLR.cc/2018/Conference — Accept (Poster)_

### Official Review · AnonReviewer1 · 2017-11-26
**This paper shows that under certain conditions, SGD learns a single convolutional filter. The conditions are a bit hard to understand and might be fairly strong.**

**Rating:** 6
**Confidence:** 3

**Review:**

This paper studies the problem of learning a single convolutional filter using SGD. The main result is: if the "patches" of the convolution are sufficiently aligned with each other, then SGD with a random initialization can recover the ground-truth parameter of a convolutional filter (single filter, ReLU, average pooling). The convergence rate, and how "sufficiently aligned" depend on some quantities related to the underlying data distribution. A major strength of the result is that it can work for general continuous distributions and does not really rely on the input distribution being Gaussian; the main weakness is that some of the distribution dependent quantities are not very intuitive, and the alignment requirement might be very high.

Detailed comments:
1. It would be good to clarify what the angle requirement means on page 2. It says the angle between Z_i, Z_j is at most \rho, is this for any i,j? From the later part it seems that each Z_i should be \rho close to the average, which would imply pairwise closeness (with some constant factor).
2. The paper first proves result for a single neuron, which is a clean result. It would be interesting to see what are values of \gamma(\phi) and L(\phi) for some distributions (e.g. Gaussian, uniform in hypercube, etc.) to give more intuitions.
3. The convergence rate depends on \gamma(\phi_0), from the initialization, \phi_0 is probably very close to \pi/2 (the closeness depend on dimension), which is  also likely to make \gamma(\phi_0) depend on dimension (this is especially true of Gaussian).
4. More precisely, \gamma(\phi_0) needs to be at least 6L_{cross} for the result to work, and L_{cross} seems to be a problem dependent constant that is not related to the dimension of the data. Also \gamma(\phi_0) depends on \gamma_{avg}(\phi_0) and \rho, when \rho is reasonable (say a constant), \gamma(\phi_0) really needs to be a constant that is independent of dimension. On the other hand, in Theorem 3.4 we can see that the upperbound on \alpha (the quality of initialization) depends on the dimension.
5. Even assuming \rho is a constant strictly smaller than \pi/2 seems a bit strong. It is certainly plausible that nearby patches are highly correlated, but what is required here is that all patches are close to the average. Given an image it is probably not too hard to find an almost all white patch and an almost all dark patch so that they cannot both be within a good angle to the average.

Overall I feel the result is interesting but hard to interpret correctly. The details of the theorem do not really support the high level claims very strongly. The paper would be much better if it goes over several example distributions and show explicitly what are the guarantees. The reviewer tried to do that for Gaussian and as I mentioned above (esp. 4) the result does not seem very impressive, maybe there are other distributions where this result works better?

After reading the response, I feel the contribution for the single neuron case does not require too much assumptions and is itself a reasonable result. I am still not convinced by the convolution case (which is the main point of this paper), as even though it does not require Gaussian input (a major plus), it still seems very far from "general distribution". Overall this is a first step in an interesting direction, so even though it is currently a bit weak I think it is OK to be accepted. I hope the revised version will clearly discuss the limitations of the approach and potential future directions as the response did.

---

> ### Author Response · Authors · 2017-12-25
> **Response**
>
> We thank the reviewer for raising these questions and insightful suggestions.
>
> To the best of our knowledge, our result is the first recovery guarantee of gradient-based algorithm for learning a ReLU activated neural network for non-Gaussian input. We acknowledge that this is just the first step toward understanding why randomly initialized (stochastic) gradient descent can learn a convolutional neural network and it is by no means a full characterization of the necessary and sufficient conditions of input distribution that lead to success of SGD/GD.
>
> At least for the one-neuron model, our proposed assumptions are general enough for explaining the success of randomly initialized gradient descent. Further, since we only deal with the single filter setting and do not take over-parametrization or other tricks into consideration that might help optimization, our proposed conditions may be further relaxed.
>
> For the convolutional filter, our key assumptions are Assumption 3.1 and Assumption 3.2. We have listed some examples in Section 3.1.
>
>
> For Detailed comments:
> 1 & 5: We would like to emphasize that this Z_i, Z_j's angle < \rho for some small \rho is one sufficient condition to secure Assumption 3.2. We add this example to emphasize the closeness of patches implies the success of learning. There are definitely infinitely more distributions satisfy Assumption 3.1 and Assumption 3.2. Further note that the equation in Assumption 3.2 is in population sense. This means that a single peak value in the patch will not affect the bound too much because it will be averaged out. Therefore, the example raised by Reviewer 1 (all white and one dark example) should still satisfy our assumption.
>
> 2. Thank for your suggestion. We have replaced figure 2(b) with the corresponding L(\phi) and gamma(\phi) for Gaussian distribution.
>
> 3. We do agree \phi_0 may be close to \pi/2 (in fact we believe \cos \phi_0 is in the order of 1/\sqrt{p} by random matrix theory) and  L_\cross may be a problem-dependent vector. However, we want to emphasize that typically, filter size is small, like 2x2 or 5x5 so even if it is dimension-dependent, requiring that \gamma(\phi) > 6L_\cross is not a strong assumption.

---

> > ### Comment · AnonReviewer1 · 2017-12-29
> > **response**
> >
> > Thanks for the response. It clears up some of my concerns. On the other hand, I do still feel some of the assumptions are a bit strong (especially in the convolution case). It is true that previously there were even no analysis for the single neuron case though, so I'm adjusting my score to a bit higher.

---

> > > ### Author Response · Authors · 2018-01-02
> > > **Thanks**
> > >
> > > Thanks for changing your score.
> > > We will definitely add more discussion about our analysis and list more future directions.

---

### Official Review · AnonReviewer2 · 2017-11-27
**Compared with relevant existing literature in this line of work, this paper extends the commonly-used Gaussian distribution assumption to a more general angular smoothness assumption, which covers a wider family of input distributions. This work is very solid in that the authors rigorously demonstrate that SGD can learn the convolutional filter in polynomial time with random initialization.**

**Rating:** 9
**Confidence:** 4

**Review:**

(a) Significance
This is an interesting theoretical deep learning paper, where the authors try to provide the theoretical insights why SGD can learn the neural network well. The motivation is well-justified and clearly presented in the introduction and related work section. And the major contribution of this work is the generalization to the non-Gaussian case, which is more in line with the real world settings. Indeed, this is the first work analyzing the input distribution beyond Gaussian, which might be an important work towards understanding the empirical success of deep learning.

(b) Originality
The division of the input space and the analytical formulation of the gradient are interesting, which are also essential for the convergence analysis. Also, the analysis framework relies on novel but reasonable distribution assumptions, and is different from the relevant literature, i.e., Li & Yuan 2017, Soltanolkotabi 2017, Zhong et al. 2017. I curious whether the angular smoothness assumptions can be applied to a more general network architecture, say two-layer neural network.

(c) Clarity & Quality
Overall, this is a well-written paper. The theoretical results are well-presented and followed by insightful explanations or remarks. And the experiments are demonstrated to justify the theoretical findings as well. The authors did a really good job in explaining the intuitions behind the imposed assumptions and justifying them based on the theoretical and experimental results. I think the quality of this work is above the acceptance bar of ICLR and it should be published in ICLR 2018.

Minor comments:
1. Figure 3 looks a little small. It is better to make them clearer.
2. In the appendix, ZZ^{\top} and the indicator function are missing in the first equation of page 13.

---

> ### Author Response · Authors · 2017-12-25
> **Response**
>
> We thank the encouraging review.
>
> We believe the angular smoothness assumptions with proper modifications can be applied to deeper models with ReLU activation since ReLU activation is very related to half-spaces.
>
> We have made the figure larger and fixed the typos. Thanks for pointing out!

---

### Official Review · AnonReviewer4 · 2017-12-16
**Interesting poly-time convergence  results under significantly more general assumptions than previous work**

**Rating:** 8
**Confidence:** 3

**Review:**

This paper considers the convergence of (stochastic) gradient descent for learning a convolutional filter with ReLU activations. It doesn't assume the input is Gaussian as in most previous work and shows that starting from random initialization, the (stochastic) gradient descent can learn the underlying convolutional filter in polynomial time. It is also shown that the convergence rate depends on the smoothness of the input distribution and the closeness of the patches.

The main contribution and the most intriguing part is that the result doesn't require assuming the input is Gaussian. Also, the guarantee holds for random initialization. The analysis that achieves these results can potentially provide better techniques for analyzing more general deep learning optimizations.

The main drawback is that the assumptions are somewhat difficult to interpret, though significantly more general than those made in previous work. It will be great if more explanations/comments are provided for these assumptions. It will be even better if one can get a simplified set of assumptions.

The presentation is clear but can be improved. Especially, more remarks would help readers to understand the paper.

minor:
-- Thm 2.1: what are w_1 w_2 here?
-- Assumption 3.1: the statement seems incomplete. I guess it should be "max_... \lambda_max(...) \geq \beta for some beta > 0"?
-- Just before Section 2.1: " This is also consistent with empirical evidence in which more data are helpful for optimization."
I don't see any evidence that more data help the optimization by filling in the holds in the distribution; they may help for other reasons. This statement here is not rigorous.

---

> ### Author Response · Authors · 2017-12-25
> **Response**
>
> We thank for your suggestions.
>
> We have added some more explanations and remarks after our assumptions and theorems in our modified paper. We hope these can help readers understand our paper better.
>
>
> For minor comments:
> -- Thm 2.1: what are w_1 w_2 here?
>
> This is a typo, we have fixed the theorem. w_2 should be w_* and w_1 should be any w such that $\theta(w,w_*) < \pi$.
>
>
> -- Assumption 3.1: the statement seems incomplete. I guess it should be "max_... \lambda_max(...) \geq \beta for some beta > 0"?
>
> Thanks for pointing out. This is a typo and we have fixed the theorem.
>
>
> -- Just before Section 2.1: " This is also consistent with empirical evidence in which more data are helpful for optimization."
> I don't see any evidence that more data help the optimization by filling in the holds in the distribution; they may help for other reasons. This statement here is not rigorous.
>
> What we mean is when minimizing the empirical loss, more data may lead to a bigger least eigenvalue of A_{w,w_*}. We agree this non-rigorous statement may lead to confusion and we have deleted it in our modified version.

---

### Decision · Program_Chairs · 2018-01-29
**ICLR 2018 Conference Acceptance Decision**

**Decision:**

Accept (Poster)

**Comment:**

Dear authors,

The reviewers all appreciated your work and agree that this a very good first step in an interesting direction.